# Efficient and Accurate Optimal Transport with Mirror Descent and Conjugate Gradients

**Mete Kemertas**  *kemertas@cs.toronto.edu*
*University of Toronto, Department of Computer Science*
*Vector Institute*

**Allan D. Jepson**  *jepson@cs.toronto.edu*
*University of Toronto, Department of Computer Science*

**Amir-massoud Farahmand**  *farahmand@mila.quebec*
*Polytechnique Montréal*
*Mila - Quebec AI Institute*
*University of Toronto, Department of Computer Science*

**Reviewed on OpenReview:** *https://openreview.net/forum?id=FVFqrxeF8e*

## Abstract

We propose *Mirror Descent Optimal Transport* (MDOT), a novel method for solving discrete optimal transport (OT) problems with high precision, by unifying temperature annealing in entropic-regularized OT (EOT) with mirror descent techniques. In this framework, temperature annealing produces a sequence of EOT dual problems, whose solution gradually gets closer to the solution of the original OT problem. We solve each problem efficiently using a GPU-parallel nonlinear conjugate gradients algorithm (PNCG) that outperforms traditional Sinkhorn iterations under weak regularization. Moreover, our investigation also reveals that the theoretical convergence rate of Sinkhorn iterations can exceed existing non-asymptotic bounds when its stopping criterion is tuned in a manner analogous to MDOT.

Our comprehensive ablation studies of MDOT-PNCG affirm its robustness across a wide range of algorithmic parameters. Benchmarking on 24 problem sets of size $n = 4096$ in a GPU environment demonstrate that our method attains high-precision, feasible solutions significantly faster than a representative set of existing OT solvers—including accelerated gradient methods and advanced Sinkhorn variants—in both wall-clock time and number of operations. Empirical convergence rates range between $O(n^2\varepsilon^{-1/4})$ and $O(n^2\varepsilon^{-1})$, where $\varepsilon$ is the optimality gap. For problem sizes up to $n = 16\,384$, the empirical runtime scales as $\widetilde{O}(n^2)$ for moderate precision and as $\widetilde{O}(n^{5/2})$ at worst for high precision. These findings establish MDOT-PNCG as a compelling alternative to current OT solvers, particularly in challenging weak-regularization regimes.

## 1 INTRODUCTION

When a statistical distance is required for an event space equipped with a metric, optimal transport (OT) distances, such as the Wasserstein metric, provide an intuitive means to account for the inherent structure of the metric space. Consequently, fast, scalable, and accurate computation of OT distances is a major problem encountered in various scientific fields. Example application areas include point cloud registration (Shen et al., 2021), color transfer (Pitie et al., 2005; Ferradans et al., 2014; Rabin et al., 2014), shape matching (Feydy et al., 2017), texture mixing (Ferradans et al., 2013; Bonneel et al., 2015) and meshing (Digne et al., 2014) in computer vision and graphics, quantum mechanics (Léonard, 2012), astronomy (Frisch et al., 2002; Levy et al., 2021) and quantum chemistry (Bokanowski & Grébert, 1996) in physics, and generative modeling (Gulrajani et al., 2017; Genevay et al., 2018), reinforcement learning (Ferns et al., 2004; Dadashi et al., 2021),

and neural architecture search (Kandasamy et al., 2018) in machine learning. Exact solvers for the discrete OT problem encounter significant computational hurdles in high dimensions, with theoretical complexity $\widetilde{O}(n^{5/2})$ and practical complexity $\widetilde{O}(n^3)$ (Lee & Sidford, 2014; Pele & Werman, 2009).

Entropic regularization, as pioneered by Cuturi (2013), has mitigated challenges in scalability by regularizing the classical problem, thereby allowing approximate solutions in $\widetilde{O}(n^2)$ time via the Sinkhorn-Knopp (SK) matrix scaling algorithm. This advancement, together with GPU parallelization, has yielded substantial speed improvements, making it several orders of magnitude faster than conventional CPU-based solvers (e.g., linear programming) in high dimensions (Peyré et al., 2019). However, these methods necessitate a delicate balance between regularization strength and convergence speed, a trade-off that can compromise the precision of the solution. Despite significant progress in recent years, many state-of-the-art solvers still struggle to strike a better trade-off than aggressively tuned Sinkhorn iterations in practice (Dvurechensky et al., 2018; Jambulapati et al., 2019; Lin et al., 2019). Although they offer superior theoretical guarantees, their practical performance is often less compelling, particularly in terms of speed and scalability. Existing algorithms either suffer from high computational complexity or do not take advantage of modern hardware capabilities, such as GPU parallelization (Tang et al., 2024). To understand and combat these challenges, we make the following contributions:

1. We empirically show that in a GPU environment the decades-old Sinkhorn-Knopp algorithm for OT can still outperform many theoretically grounded recent OT algorithms in practice, especially when tuned with a seemingly unconventional stopping criterion formula proposed here (Fig. 5).

2. We introduce mirror descent optimal transport (MDOT), a method which generalizes temperature annealing in entropic OT (EOT) (Schmitzer, 2019; Feydy, 2020), and connects temperature annealing to mirror descent (Alg. 1).

3. We introduce an instantiation of MDOT that empirically improves speed and robustness to temperature (regularization strength) decay rate compared to $\varepsilon$-scaling of Schmitzer (2019) (Fig. 2).

4. We show that MDOT can compute high precision, feasible solutions and its performance can be boosted by adopting a specialized GPU-parallel conjugate gradients (CG) algorithm developed here (Alg. 2); this method is highly competitive in practice, as we show empirically (Figs. 5, 6, 10-19).

The remainder of this paper is organized as follows. In the next section, we introduce our notation and the necessary background, followed by related work in Sec. 3. In Sec. 4.1-4.2, we introduce the MDOT framework and establish its connection to temperature annealing strategies, and make some practical recommendations. In Sec. 4.3, we introduce the non-linear CG algorithm to be used within MDOT as an alternative to SK. In Sec. 5, we benchmark various algorithms on upsampled MNIST ($n = 4096$) under $L_1$ and squared $L_2$ costs, and a color transfer problem *in terms of wall-clock time*, and further study the operation count dependence of the proposed algorithm on problem size $n$. Lastly, we present concluding remarks in Sec. 6.

## 2 Background

Here, we present our notation, the basics of EOT, and the necessary background on mirror descent and CG.

**Notation and Definitions.** We consider discrete OT, where the event space is finite with $n$ particles and $\Delta_n \subset \mathbb{R}_{\geq 0}^n$ is the $(n-1)$-simplex. The row sum of an $n \times n$ matrix $P$ is $\boldsymbol{r}(P) \coloneqq P\mathbf{1}$ and the column sum is $\boldsymbol{c}(P) \coloneqq P^\top \mathbf{1}$. Given marginals $\boldsymbol{r}, \boldsymbol{c} \in \Delta_n$, the transportation polytope is written as $\mathcal{U}(\boldsymbol{r}, \boldsymbol{c}) = \{P \in \mathbb{R}_{\geq 0}^{n \times n} \mid \boldsymbol{r}(P) = \boldsymbol{r}, \boldsymbol{c}(P) = \boldsymbol{c}\}$. Division, exp and log over vectors or matrices are element-wise. Vectors in $\mathbb{R}^n$ are column vectors, and $(\boldsymbol{x}, \boldsymbol{y})$ denotes the concatenation of $\boldsymbol{x}$ and $\boldsymbol{y}$. Vector and Frobenius inner products alike are given by $\langle \cdot, \cdot \rangle$. Hadamard product is given by $A \odot B$. An $n \times n$ diagonal matrix with $\boldsymbol{x} \in \mathbb{R}^n$ along the diagonal is written as $\mathbf{D}(\boldsymbol{x})$, and the vector formed by the diagonal entries of a matrix $Q$ is $\mathbf{diag}(Q)$. LogSumExp reductions over the rows and columns of $X \in \mathbb{R}^{n \times n}$ are given by $\mathrm{LSE}_r(X) \coloneqq \log\big(\exp\{X\}\mathbf{1}\big)$ and $\mathrm{LSE}_c(X) \coloneqq \log\big(\exp\{X^\top\}\mathbf{1}\big)$. The Shannon entropy of $\boldsymbol{r} \in \Delta_n$ is denoted $H(\boldsymbol{r}) = -\langle \boldsymbol{r}, \log \boldsymbol{r} \rangle$ with the convention that $0 \cdot \log 0 = 0$. Under the same convention, the KL divergence $D_{\mathrm{KL}}(\boldsymbol{r}|\boldsymbol{r}') = \langle \boldsymbol{r}, \log(\boldsymbol{r}/\boldsymbol{r}') \rangle + \langle \boldsymbol{r}' - \boldsymbol{r}, \mathbf{1} \rangle$ for $\boldsymbol{r}, \boldsymbol{r}' \in \mathbb{R}_{\geq 0}^n$ given $\boldsymbol{r}$ absolutely continuous with respect to $\boldsymbol{r}'$.

## 2.1 Optimal Transport

Given a cost matrix $C \in [0,1]^{n \times n}$, where $C_{ij}$ is the transportation cost between the $i^{\text{th}}$ and $j^{\text{th}}$ particles, we study the EOT problem:

$$\underset{P \in \mathcal{U}(\boldsymbol{r}, \boldsymbol{c})}{\text{minimize}} \quad \langle P, C \rangle - \frac{1}{\gamma} H(P), \tag{1}$$

where $\gamma > 0$. Here, the regularization weight $\gamma^{-1}$ is called *temperature*. The Lagrangian of (1) is strictly convex in $P$, which renders the solution $P^*(\gamma)$ unique. $P^*(\gamma)$ converges to a solution of the unregularized OT problem as $\gamma \to \infty$ and admits the following form (Cuturi, 2013):

$$P_{ij}(\boldsymbol{u}, \boldsymbol{v}; \gamma) = \exp\{u_i + v_j - \gamma C_{ij}\}, \tag{2}$$

where $\boldsymbol{u}, \boldsymbol{v} \in \mathbb{R}^n$. An optimal pair $(\boldsymbol{u}, \boldsymbol{v})$ minimizes the following convex dual problem (Lin et al., 2019):

$$\underset{\boldsymbol{u}, \boldsymbol{v} \in \mathbb{R}^n}{\text{minimize}} \quad g(\boldsymbol{u}, \boldsymbol{v}; \gamma, \boldsymbol{r}, \boldsymbol{c}) = \sum_{ij} P_{ij}(\boldsymbol{u}, \boldsymbol{v}; \gamma) - \langle \boldsymbol{u}, \boldsymbol{r} \rangle - \langle \boldsymbol{v}, \boldsymbol{c} \rangle, \tag{3}$$

where $\nabla_{\boldsymbol{u}} g = \boldsymbol{r}(P) - \boldsymbol{r}$ and $\nabla_{\boldsymbol{v}} g = \boldsymbol{c}(P) - \boldsymbol{c}$. The gradient norm naturally measures the constraint violation; $L_1$ norm is typically used to monitor convergence as it relates to the total variation metric between probability distributions (Altschuler et al., 2017). The Sinkhorn-Knopp (SK) algorithm (see Alg. 4 in Appx. A) can be used to minimize $g$ with guaranteed convergence to dual-optimal variables as the number of iterations $k \to \infty$ (Sinkhorn & Knopp, 1967; Sinkhorn, 1967; Franklin & Lorenz, 1989; Knight, 2008). Dvurechensky et al. (2018) showed that SK can be used to compute a solution $P \in \mathcal{U}(\boldsymbol{r}, \boldsymbol{c})$ satisfying $\langle P - P^*, C \rangle \leq \varepsilon$ with complexity $\widetilde{O}(n^2/\varepsilon^2)$, where $P^*$ is an optimal solution of the unregularized OT problem. In particular, one first minimizes the dual objective until the $L_1$ norm of its gradient is below a prescribed threshold, then applies the rounding algorithm of Altschuler et al. (2017) on the infeasible plan given by (2) to obtain $P \in \mathcal{U}(\boldsymbol{r}, \boldsymbol{c})$ with an upper bound on the primal cost increase.

## 2.2 Mirror Descent

Consider a constrained convex minimization problem $\min_{P \in \mathcal{F}} f(P)$, where the convex objective $f : \mathcal{D} \to \mathbb{R}$ is defined over some domain $\mathcal{D}$, and the feasible set $\mathcal{F} \subseteq \mathcal{D}$, e.g., in OT, we take $f(P) = \langle P, C \rangle$, $\mathcal{F} = \mathcal{U}(\boldsymbol{r}, \boldsymbol{c})$ and $\mathcal{D} = \mathbb{R}_{\geq 0}^{n \times n}$. Mirror descent (MD) method of Nemirovski & Yudin (1983) generalizes *projected* gradient descent (GD) by first choosing a strictly convex function $h : \mathcal{D} \to \mathbb{R}$ called the *mirror map*, which induces a *Bregman divergence* $D_h(P|Q) \geq 0$ between points $P, Q \in \mathcal{D}$ as the difference between $h(P)$ and its first order approximation around $Q$:

$$D_h(P|Q) = h(P) - h(Q) - \langle \nabla h(Q), P - Q \rangle. \tag{4}$$

Then, the variable $P^{(t)}$ is updated as follows (Bubeck, 2015):

$$\hat{P}^{(t+1)} = \nabla h^{-1} \left( \nabla h(P^{(t)}) - \Delta^{(t)} \nabla f(P^{(t)}) \right) \quad \text{(5a)} \qquad P^{(t+1)} = \underset{P \in \mathcal{F} \cap \mathcal{D}}{\arg\min} D_h(P|\hat{P}^{(t+1)}) \quad \text{(5b)}$$

where $\Delta^{(t)} > 0$ is the step size. For $h(P) = \|P\|_2^2/2$, projected GD is recovered as a special case since $\nabla h(P) = P$ and $D_h(P|Q) = \|P - Q\|_2^2/2$. For certain problem geometries (e.g., if $\mathcal{D}$ is the simplex), choosing a different $h$ provably accelerates convergence. The idea is that in (5a), $\nabla h : \mathcal{D} \to \mathcal{D}^*$ maps primal variables $P$ to a dual space $\mathcal{D}^*$, where applying a gradient update better aligns with the local curvature of the underlying geometry (Bubeck, 2015). Once the updated point is mapped back to primal space $\mathcal{D}$ via $\nabla h^{-1} : \mathcal{D}^* \to \mathcal{D}$, (5b) performs a *Bregman projection* onto the feasible set $\mathcal{F}$. Plugging (5a) into (5b) and rearranging yields an equivalent form with another interpretation (Beck & Teboulle, 2003):

$$P^{(t+1)} = \underset{P \in \mathcal{F} \cap \mathcal{D}}{\arg\min} \{ \langle \nabla_P f(P^{(t)}), P \rangle + \frac{1}{\Delta^{(t)}} D_h(P|P^{(t)}) \}. \tag{6}$$

Each MD step (6) seeks an update jointly maximizing the inner product with the steepest descent direction, $-\nabla_P f(P^{(t)})$, subject to (i) a penalty for *diverging* from the current point $P^{(t)}$ and (ii) feasibility constraints.

A common choice is the negative entropy mirror map $h(P) = \langle P, \log P \rangle$ in domain $\mathcal{D} = \mathbb{R}_{\geq 0}^{n \times n}$, which yields $D_h(P|Q) = D_{\text{KL}}(P|Q)$. Throughout this work, we only use this mirror map.[1] Hence, we write (5) explicitly

---

[1]See Di Marino & Gerolin (2020) for other divergence-regularized optimal transport problems (using mirror maps besides negative Shannon entropy). They consider a single step of (6), while we focus on multiple steps.

for this case, where $\nabla h(P) = \mathbf{1} + \log P$ and $\nabla h^{-1}(R) = \exp\{R - \mathbf{1}\}$,

$$\hat{P}^{(t+1)} = P^{(t)} \odot \exp\{-\Delta^{(t)}\nabla f(P^{(t)})\} \qquad \text{(7a)} \qquad P^{(t+1)} = \underset{P \in \mathcal{F} \cap \mathcal{D}}{\arg\min}\, D_{\mathrm{KL}}(P|\hat{P}^{(t+1)}). \qquad \text{(7b)}$$

Here, we re-weight $P^{(t)}$ entrywise by a factor of $\exp\{-\Delta^{(t)}\nabla f(P^{(t)})\}$, followed by a KL projection onto $\mathcal{F}$, e.g., if $\mathcal{F}$ is the probability simplex, (7b) is a simple renormalization $P \mapsto P/\|P\|_1$. We leverage the properties of this mirror map for the case $\mathcal{F} = \mathcal{U}(\boldsymbol{r}, \boldsymbol{c})$ to design MDOT in Sec. 4.

## 3 Related Work

Acceleration of approximate OT solvers has been a focus of machine learning research since the seminal work of Cuturi (2013). For instance, Altschuler et al. (2017) proposed the Greenkhorn algorithm, which greedily selects individual rows or columns to scale at a given step and requires fewer row/column updates than SK to converge, but performs poorly due to low GPU utilization unless $n$ is extremely large. Dvurechensky et al. (2018) proposed an adaptive primal-dual accelerated gradient descent (APDAGD) algorithm. Lin et al. (2019) later proposed adaptive primal-dual accelerated *mirror* descent (APDAMD) with theoretical guarantees. Lin et al. (2019) showed APDAMD to outperform APDAGD *in terms of number of iterations*, but not SK. Further, these tests only covered a high relative error regime ($>50\%$); we investigate a broader scope down to $10^{-9}$ error in Section 5. Modest gains over SK in terms of number of iterations in the same regime were later obtained by Lin et al. (2022) via an accelerated alternating minimization (AAM) algorithm similar to that of Guminov et al. (2021). Notably, APDAMD applies mirror descent to the dual (3) of the EOT problem, while we apply it to the primal of the unregularized OT problem, i.e., problem (1) as $\gamma \to \infty$.

Application of mirror descent to the primal of the OT problem has also been considered. Yang & Toh (2022) propose a more general inexact mirror descent algorithm (iBPPA), which they apply to the OT problem. Our approach differs from theirs in several ways, most notably in how we decide to terminate the Bregman projection inner loop to solve (7b) inexactly (details in Section 4.2). Recently, Ballu & Berthet (2023) introduced Mirror Sinkhorn (MSK), which also takes gradient steps in the dual space as in (7a), but instead of approximately projecting onto $\mathcal{U}(\boldsymbol{r}, \boldsymbol{c})$ as in (7b) (as we do here), they alternately project onto $\mathcal{U}(\cdot, \boldsymbol{c})$ and $\mathcal{U}(\boldsymbol{r}, \cdot)$ via Sinkhorn updates, satisfying only half of the marginal constraints at a time. Our experiments in Sec. 5 suggest that this approach is efficient only in the low precision regime. Furthermore, MSK requires maintaining a running average of the transport plan at each iteration, precluding a straightforward $O(n)$ memory implementation. In contrast, all algorithms presented here admit $O(n)$ memory implementations, assuming individual cost matrix entries can be computed on-the-fly in $O(1)$ time. Xie et al. (2020) previously proposed an algorithm (IPOT) similar to MSK with a fixed, even number of Sinkhorn updates (usually 2) following temperature updates. Alg. 3.5 of Feydy (2020) is also similar to these algorithms in spirit and is discussed thoroughly in Sec. 5.2. As discussed in detail in Sec. 4.1, well-known $\varepsilon$-scaling strategies are also closely related (Kosowsky & Yuille, 1994; Schmitzer, 2019). Similar ideas have also been applied to the Wasserstein barycenter problem (Gramfort et al., 2015; Xie et al., 2020), which is left outside the scope of this work.

An alternative line of acceleration research focuses on multi-scale strategies, which employ clustering or grid-based methods to solve a series of coarse-to-fine OT problems and are sometimes combined with $\varepsilon$-scaling (Schmitzer, 2016; 2019; Feydy, 2020). These are known to provide performance gains when the marginals are defined over well-clustered particles or in low-dimensional event spaces (Peyré et al., 2019). Lastly, in a similar spirit to our use of non-linear CG here, curvature-aware convex optimization techniques such as L-BFGS have also been considered for OT, e.g., Mérigot (2011); Blondel et al. (2018); however, scalability, precision and better performance than SK on GPUs has not been demonstrated simultaneously to our knowledge. Tang et al. (2024) recently adopted Newton's method with Hessian sparsification to efficiently use second order information, but their key sparsification strategy is maximally utilized only on CPUs.

## 4 A Mirror Descent Framework for Optimal Transport

### 4.1 Temperature Annealing as Mirror Descent

The OT objective $\langle P, C \rangle$ has a constant gradient $\nabla_P \langle P, C \rangle = C$. Given step sizes $\Delta^{(t)} > 0$ at time $t \geq 0$, we obtain mirror descent iterates (using the negative entropy mirror map) from (7)

$$P^{(t+1)} = \underset{P \in \mathcal{U}(\boldsymbol{r}, \boldsymbol{c})}{\arg \min} D_{\mathrm{KL}}\Big(P|P^{(t)} \odot \exp\{-\Delta^{(t)}C\}\Big). \tag{8}$$

Comparing the conditioning term $P^{(t)} \odot \exp\{-\Delta^{(t)}C\}$ above with the Lagrangian closed form solution to the EOT problem in (2), namely $P(\boldsymbol{u}, \boldsymbol{v}; \gamma) = P(\boldsymbol{u}\boldsymbol{1}^\top + \boldsymbol{1}\boldsymbol{v}^\top - \gamma C)$, suggests considering an MD step starting with a $P^{(t)}$ of this same form. Specifically, we find the following result (proved in Appx. A.1).

**Lemma 4.1.** *Given any initial $\boldsymbol{u}, \boldsymbol{v} \in \mathbb{R}^n$, $\gamma \geq 0$ and $\Delta_\gamma > 0$, we have*

$$P(\boldsymbol{u}^*, \boldsymbol{v}^*; \gamma + \Delta_\gamma) = \underset{P \in \mathcal{U}(\boldsymbol{r}, \boldsymbol{c})}{\arg \min} D_{\mathrm{KL}}\Big(P|P(\boldsymbol{u}, \boldsymbol{v}; \gamma) \odot \exp\{-\Delta_\gamma C\}\Big), \tag{9}$$

*where $\boldsymbol{u}^*, \boldsymbol{v}^* \in \arg\min g(\gamma + \Delta_\gamma, \boldsymbol{r}, \boldsymbol{c})$.*

That is, if we begin an MD step for the OT problem with any $P^{(t)} = P(\boldsymbol{u}, \boldsymbol{v}; \gamma^{(t)})$ (i.e., in the Lagrangian form but not necessarily optimal for the EOT problem), then the next MD iterate is the unique solution to the EOT dual problem at $\gamma^{(t+1)} = \gamma^{(t)} + \Delta_\gamma^{(t)}$, namely $P^{(t+1)} = P^*(\gamma^{(t+1)})$. Therefore, by induction, if we begin the MD iteration (8) for the OT problem with such an initial $P^{(0)}$, then the MD iterates $P^{(t)}$ trace solutions of the EOT problems at increasing values of $\gamma^{(t)}$.

For initialization, recall that the EOT solution at $\gamma=0$ is the maximum entropy coupling, $\boldsymbol{r}\boldsymbol{c}^\top = \lim_{\gamma \to 0^+} \arg\min_{P \in \mathcal{U}(\boldsymbol{r}, \boldsymbol{c})} \langle P, C \rangle - \frac{1}{\gamma}H(P)$. Thus, we can initialize $\gamma^{(0)} = 0$ and $P^{(0)} = P(\log \boldsymbol{r}, \log \boldsymbol{c}; 0) = \boldsymbol{r}\boldsymbol{c}^\top$. Alternatively, in view of Lemma 4.1, it would suffice to use $P^{(0)} = P(\boldsymbol{u}, \boldsymbol{v}; 0)$ for any $\boldsymbol{u}, \boldsymbol{v} \in \mathbb{R}^n$ at $\gamma^{(0)} = 0$. That is, $P^{(0)}$ can be any rank-1 matrix in $\mathbb{R}_{>0}^{n \times n}$.

Finally, note that for the purpose of computing $P^{(t+1)}$ in (8), Lemma 4.1 ensures that any values $\boldsymbol{u}^{(t)}$ and $\boldsymbol{v}^{(t)}$ can serve to provide a suitable $P^{(t)} = P(\boldsymbol{u}^{(t)}, \boldsymbol{v}^{(t)}; \gamma^{(t)})$. Therefore, when implementing the MD iteration in (8) we do not need to converge to exact optimality each step (see Fig. 1). We formalize this discussion and provide further insight in the following proposition.

**Proposition 4.2.** *Let $\gamma^{(0)} = 0$ and $\gamma^{(t+1)} = \gamma^{(t)} + \Delta_\gamma^{(t)}$, which together imply $\gamma^{(t+1)} = \sum_{t'=0}^{t} \Delta_\gamma^{(t')}$. Suppose $P^{(0)} \in \mathbb{R}_{>0}^{n \times n}$ is rank-1 and $P^{(t)}$ are computed via (8) for $t \geq 0$. The following are true.*

1. *$P^{(t+1)} = P^*(\gamma^{(t+1)})$, i.e., the solution of the EOT problem (1) at $\gamma^{(t+1)}$.*

2. *Given any $\boldsymbol{u}, \boldsymbol{v} \in \mathbb{R}^n$, $P^{(t+1)} = \arg\min_{P \in \mathcal{U}(\boldsymbol{r}, \boldsymbol{c})} D_{\mathrm{KL}}(P|P(\boldsymbol{u}, \boldsymbol{v}; \gamma^{(t+1)}))$.*

3. *$\langle P^{(t)} - P^*, C \rangle \leq H_{\min}(\boldsymbol{r}, \boldsymbol{c})/\gamma^{(t)}$, where $P^* \in \arg\min_{P \in \mathcal{U}(\boldsymbol{r}, \boldsymbol{c})} \langle P, C \rangle$ and $H_{\min}(\boldsymbol{r}, \boldsymbol{c}) := \min\big(H(\boldsymbol{r}), H(\boldsymbol{c})\big)$.*

4. *$\langle P^{(t)} - P^{(t+1)}, C \rangle = \frac{1}{\Delta_\gamma^{(t)}}\Big(D_{\mathrm{KL}}\big(P^{(t)}|P^{(t+1)}\big) + D_{\mathrm{KL}}\big(P^{(t+1)}|P^{(t)}\big)\Big)$ for all $t \geq 0$.*

A detailed proof is deferred to Appx. A.2. Next we comment on each statement in the order presented.

**1.** proves the equivalence of MD iterates to a sequence of solutions to EOT problems with decreasing temperature, and therefore connects MD to temperature annealing.

**2.** shows that MD iterates $P^{(t+1)}$ derived from a rank-1 initialization can be written independently of the previous solution $P^{(t)}$ (cf. 8), but rather only as a function of $\gamma^{(t+1)}$. This means that we need not start from an exact minimizer of the previous KL projection problem to reach the correct solution of the new problem. Given this result and Lemma 4.1, MD reduces in the dual space to a numerical continuation method (Allgower & Georg, 2003), in which we numerically trace the curve $\boldsymbol{u}^*(\gamma), \boldsymbol{v}^*(\gamma)$:[2]

$$\boldsymbol{u}^{(t)}, \boldsymbol{v}^{(t)} \approx \underset{\boldsymbol{u}, \boldsymbol{v} \in \mathbb{R}^n}{\arg \min} g(\boldsymbol{u}, \boldsymbol{v}; \gamma^{(t)}, \boldsymbol{r}, \boldsymbol{c}), \tag{10}$$

where each problem is initialized with some guess $\boldsymbol{u}', \boldsymbol{v}' \in \mathbb{R}^n$. This is the key idea behind MDOT.

**3.** bounds the primal optimality gap at a given step $t \geq 1$ in terms of the entropies of the marginals $\boldsymbol{r}, \boldsymbol{c}$. Since $\min(H(\boldsymbol{r}), H(\boldsymbol{c})) \leq \log n$ for $\boldsymbol{r}, \boldsymbol{c}$ on the $(n-1)$-simplex, this is a tighter bound than the standard upper bound $\gamma^{-1} \log n$ used in prior work (Altschuler et al., 2017; Dvurechensky et al., 2018; Lin et al., 2019).[3]

---

[2]Slight abuse of terminology: in fact, there is a space of optimal curves, since $g(\boldsymbol{u}, \boldsymbol{v}; \gamma) = g(\boldsymbol{u}+\delta\boldsymbol{1}, \boldsymbol{v}-\delta\boldsymbol{1}; \gamma)$ for any finite $\delta$.

[3]This $\log n$ term appears in the time complexity of various algorithms, but is hidden in $\widetilde{O}$-notation.

**Visualization of MDOT in Dual Space**

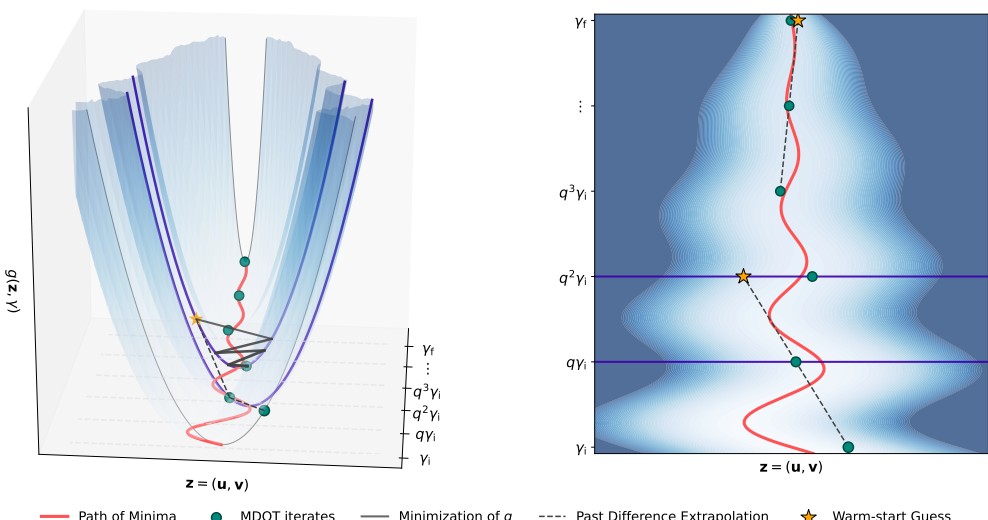

Figure 1: In each iteration of Alg. 1, MDOT approximately minimizes $g(\boldsymbol{u}, \boldsymbol{v}; \gamma)$ as in (10) to compute duals $(\boldsymbol{u}^{(t)}, \boldsymbol{v}^{(t)})$ (i.e., "MDOT iterates"), starting initially with $\gamma_i$ and increasing $\gamma$ by a factor of $q > 1$ per iteration until a final value $\gamma_f$. With increasing $\gamma$ (weaker entropic regularization), the convex dual objectives defined over $\mathbb{R}^{2n}$ become increasingly ill-conditioned (illustrated here in 1D by parabolas with increasingly high curvature) and therefore harder to solve. At each step, MDOT uses approximate solutions of prior problems to produce an initial guess for the next problem **(left)**, and generates increasingly accurate warm-start guesses to mitigate the difficulty posed by ill-conditioning **(right)**.

*4.* shows the one-step reduction in transport cost *with equality*; this is in contrast to the more standard analysis of MD where the improvement is bounded with an inequality.

### 4.2 A mirror descent method for optimal transport: MDOT

Using insights derived in the previous section, we construct the MDOT method shown in Alg. 1. As illustrated in Fig. 1, MDOT minimizes a sequence of EOT dual objectives $g(\gamma^{(t)})$. At each step, MDOT uses prior approximate minimizers $\boldsymbol{u}^{(t-k)}, \boldsymbol{v}^{(t-k)}$ of $g(\gamma^{(t-k)})$ for $k \in [0, K]$ to extrapolate better initial guesses (warm-starting) for the next problem at $t+1$ (see Fig. 1, where $K = 1$ is depicted).

The MDOT loop is summarized with the following steps:

- **L3**: Choose stopping criterion $\varepsilon_{\mathrm{d}}$ for $\|\nabla g(\gamma, \boldsymbol{r}, \boldsymbol{c})\|_1$.
- **L4**: Obtain smoothed marginals $\tilde{\boldsymbol{r}}, \tilde{\boldsymbol{c}}$ to avoid zero or infinitesimal entries.
- **L5**: Rank-1 initial guess with positive $\tilde{\boldsymbol{r}}, \tilde{\boldsymbol{c}}$.
- **L6**: Solve (10) given initial $\boldsymbol{u}', \boldsymbol{v}'$, i.e., minimize $g(\gamma, \tilde{\boldsymbol{r}}, \tilde{\boldsymbol{c}})$ until $\|\nabla g(\gamma, \tilde{\boldsymbol{r}}, \tilde{\boldsymbol{c}})\|_1 \leq \varepsilon_{\mathrm{d}}/2$. By the triangle inequality, this implies $\|\nabla g(\gamma, \boldsymbol{r}, \boldsymbol{c})\|_1 \leq \varepsilon_{\mathrm{d}}$.
- **L7**: Exit if $\gamma$ reached user input $\gamma_f$, otherwise continue.
- **L8**: Increase $\gamma$ for the next iteration.
- **L9**: Using previous $\boldsymbol{u}^{(t-k)}, \boldsymbol{v}^{(t-k)}$ for $k \geq 0$, extrapolate a warm-start guess for the new KL projection (or, EOT dual) problem of minimizing $g(\gamma^{(t+1)}, \boldsymbol{r}, \boldsymbol{c})$.

In L13, approximation of $P^*(\gamma_f)$ is rounded via Altschuler et al. (2017) onto $\mathcal{U}(\boldsymbol{r}, \boldsymbol{c})$; see Alg. 3 in Appx. A.

---

**Algorithm 1** $\mathrm{MDOT}(C, \boldsymbol{r}, \boldsymbol{c}, \gamma_i, \gamma_f, p \geq 1, q > 1)$

1: $t \leftarrow 1$, $\gamma \leftarrow \min(\gamma_i, \gamma_f)$
2: **while** True **do**
3:     $\varepsilon_{\mathrm{d}} \leftarrow H_{\min}(\boldsymbol{r}, \boldsymbol{c})/\gamma^p$
4:     $(\tilde{\boldsymbol{r}}, \tilde{\boldsymbol{c}}) \leftarrow (1 - \frac{\varepsilon_{\mathrm{d}}}{4}) \cdot (\boldsymbol{r}, \boldsymbol{c}) + \frac{\varepsilon_{\mathrm{d}}}{4n} \cdot \mathbf{1}_{2n}$
5:     **if** $t == 1$ **then** $\boldsymbol{u}', \boldsymbol{v}' \leftarrow \log \tilde{\boldsymbol{r}}, \log \tilde{\boldsymbol{c}}$
6:     $\boldsymbol{u}^{(t)}, \boldsymbol{v}^{(t)} \leftarrow$ Given initialization $\boldsymbol{u}', \boldsymbol{v}'$ minimize $g(\gamma, \tilde{\boldsymbol{r}}, \tilde{\boldsymbol{c}})$ until $\|\nabla g\|_1 \leq \varepsilon_{\mathrm{d}}/2$
7:     **if** $\gamma == \gamma_f$ **then break**
8:     $\Delta_\gamma \leftarrow (q-1)\gamma$,   $\gamma \leftarrow \min(\gamma + \Delta_\gamma, \gamma_f)$
9:     $\boldsymbol{u}', \boldsymbol{v}' \leftarrow \mathrm{WarmStart}(\boldsymbol{u}^{(t)}, \boldsymbol{v}^{(t)}; \cdots)$
10:     $t \leftarrow t + 1$
11: **end while**
12: $P \leftarrow \exp\{\boldsymbol{u}^{(t)} \mathbf{1}_n^\top + \mathbf{1}_n \boldsymbol{v}^{(t)\top} - \gamma_f C\}$
13: Output $P \leftarrow \mathrm{Round}(P, \boldsymbol{r}, \boldsymbol{c})$

---

Using (i) the guarantee that $\|\nabla g(\gamma_f, \boldsymbol{r}, \boldsymbol{c})\|_1 \leq H_{\min}(\boldsymbol{r}, \boldsymbol{c})/\gamma_f$, (ii) the third statement of Prop. 4.2, and (iii) Lemma 7 of Altschuler et al. (2017), we obtain a guarantee on the quality of the solution.

**Remark 4.3.** *If $\gamma_{\mathrm{f}} \geq 5H_{\min}(\boldsymbol{r}, \boldsymbol{c})/2\varepsilon$, the output $P \in \mathcal{U}(\boldsymbol{r}, \boldsymbol{c})$ of Alg. 1 satisfies $\langle P - P^*, C \rangle \leq \varepsilon + \widetilde{O}(\varepsilon^2)$.*

The proof follows from a special case of Prop. 4.5, which is deferred to Sec. 4.2.2. Next, we discuss key MDOT steps listed above, how they differ from existing work, and trade-offs of input parameters.

**Stopping criterion (L3) and input $p$.** Several prior methods such as IPOT (Xie et al., 2020), Alg. 3.5 of Feydy (2020) and MSK (Ballu & Berthet, 2023) that use MD ideas do not enforce any constraints on $\|\nabla g\|_1$. Rather, they only take a few Sinkhorn steps after $\gamma$ is increased. Feydy (2020) takes increasingly bigger updates $\Delta_\gamma = (q-1)\gamma$ as we do in L8, such that when $\gamma$ is large, a few Sinkhorn steps are insufficient to prevent divergence from the optimal curve $\boldsymbol{u}^*(\gamma), \boldsymbol{v}^*(\gamma)$. This effectively caps their precision as we show in Sec. 5. IPOT takes fixed steps (typically $\Delta_\gamma = 1$), which helps trace the curve $\boldsymbol{u}^*(\gamma), \boldsymbol{v}^*(\gamma)$ closely, but reduces the regularization weight $\gamma^{-1}$ more slowly; see Appx. G for further discussion and empirical data. On the other hand, MSK takes increasingly smaller $\Delta_\gamma^{(t)} = O(1/\sqrt{t})$, such that its convergence rate suffers (shown in Sec. 5). All three rely on Sinkhorn updates, while MDOT can benefit from any (potentially better) convex optimization algorithms for minimizing $g$. The practical approach of Yang & Toh (2022) also uses a fixed $\Delta_\gamma$ and Sinkhorn iteration. As the stopping criterion, they measure $D_{\mathrm{KL}}(\mathrm{Round}(P, \boldsymbol{r}, \boldsymbol{c})|P)$ after each Sinkhorn step to satisfy an inexactness condition for the KL projection, which facilitates their theoretical analysis. This incurs some $O(n^2)$ overhead per inner loop iteration. In contrast, we only check the $L_1$ norm of the dual gradient as in L6 (which can be evaluated in $O(n)$ time during optimization), since by Proposition 4.2 maintaining the structure $P(\boldsymbol{u}, \boldsymbol{v}; \gamma)$ throughout ensures that $P^*(\gamma_{\mathrm{f}})$ can be recovered theoretically, regardless of how inexact previous KL projections were.

Another line of work runs (some version of) a single iteration of MDOT (Altschuler et al., 2017; Dvurechensky et al., 2018; Lin et al., 2019). In these works, $\varepsilon_{\mathrm{d}} \propto (\log n)/\gamma$, while we adapt it according to the entropy of $\boldsymbol{r}, \boldsymbol{c}$ using the third statement of Prop. 4.2. In Appx. C, the use of $H_{\min}(\boldsymbol{r}, \boldsymbol{c})$ instead of $\log n$ is shown to yield speedups of order $\log n/H_{\min}(\boldsymbol{r}, \boldsymbol{c})$. Our use of $p \geq 1$ is also novel and controls how closely the curve $\boldsymbol{u}^*(\gamma), \boldsymbol{v}^*(\gamma)$ is traced. We study theoretical benefits and practical trade-offs in Sec. 4.2.2.

**Marginal smoothing (L4).** Our "smoothing" of the target marginals $\boldsymbol{r}$ and $\boldsymbol{c}$ (mixing in the uniform distribution) follows prior work by Dvurechensky et al. (2018) and Lin et al. (2019). This step helps provide convergence guarantees for certain choices of minimization algorithms for the dual $g$ in L6. Since the mixing weight used is proportional to $\gamma^{-p}$, it gradually decreases with the temperature in our case. This scheme smoothes marginals more aggressively in earlier iterations of MDOT when $\gamma$ is small. In Appx. D, we empirically show the performance benefits of this variable smoothing.

**EOT dual minimization (L6).** The algorithm for minimizing $g(\gamma, \boldsymbol{r}, \boldsymbol{c})$ is left unspecified here for generality; for example, Sinkhorn iteration can be used. In Sec. 4.3, we introduce a new algorithm (PNCG).

**Temperature decay and input $q$ (L8).** L8 effectively sets $\gamma^{(t+1)} = q\gamma^{(t)}$, which follows prior work (Schmitzer, 2019; Feydy, 2020). With larger MD steps (higher $q$), the extrapolation of duals for warm-starting is over a longer range, which degrades quality and poses a trade-off, investigated in Sec. 4.2.1.

**Warm-starting duals (L9).** Our explicit warm-starting approach in Sec. 4.2.1 is novel to our knowledge and was left unspecified in Alg. 1 for generality. We show that $\varepsilon$-scaling as implemented by Schmitzer (2019) and Feydy (2020) amount to an *implicit* warm-starting, which is shown to be less effective than ours.

### 4.2.1 Warm-starting KL Projections

Assume that at each prior temperature value, we obtained the dual-optimal $\boldsymbol{u}^*(\gamma^{(t)}), \boldsymbol{v}^*(\gamma^{(t)})$ without error. How should $\boldsymbol{u}, \boldsymbol{v}$ be initialized for $\gamma^{(t+1)}$? A simple, memory-efficient approach is to consider a Taylor expansion around recent $\gamma$ to predict $\boldsymbol{u}^*(\gamma^{(t+1)}), \boldsymbol{v}^*(\gamma^{(t+1)})$. Letting $\boldsymbol{z} = (\boldsymbol{u}, \boldsymbol{v})$ to reduce clutter:

$$\boldsymbol{z}^*(\gamma^{(t+1)}) = \boldsymbol{z}^*(\gamma^{(t)}) + \frac{\partial \boldsymbol{z}^*}{\partial \gamma}(\gamma^{(t)})(\gamma^{(t+1)} - \gamma^{(t)}) + \dots \tag{11}$$

As we cannot compute $\partial \boldsymbol{z}^*/\partial \gamma$ analytically, we use a numerical approximation (backward finite differencing) $\partial \boldsymbol{z}^*(\gamma^{(t)})/\partial \gamma \approx \big(\boldsymbol{z}^*(\gamma^{(t)}) - \boldsymbol{z}^*(\gamma^{(t-1)})\big)/\Delta_\gamma^{(t-1)}$ and keep only the first two terms in (11):

$$\boldsymbol{z}^*(\gamma^{(t+1)}) \approx \boldsymbol{z}^*(\gamma^{(t)}) + \frac{\Delta_\gamma^{(t)}}{\Delta_\gamma^{(t-1)}}\Big(\boldsymbol{z}^*(\gamma^{(t)}) - \boldsymbol{z}^*(\gamma^{(t-1)})\Big). \tag{12}$$

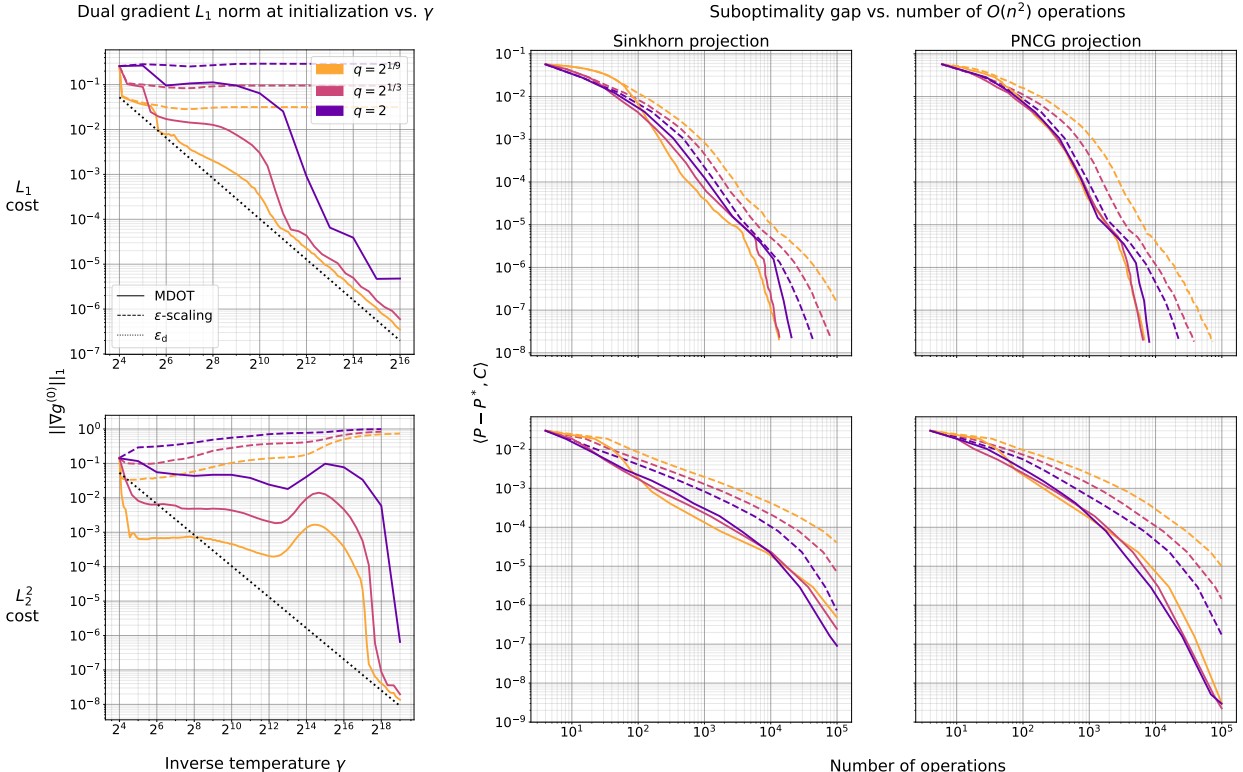

Figure 2: Comparison of the MDOT warm-start proposed in Sec. 4.2.1 to $\varepsilon$-scaling. Curves show the median over 36 upsampled-MNIST problems ($n$=4096) under $L_1$ (**top**) and $L_2^2$ (**bottom**) distance costs (see Sec. 5 for details). In all experiments, $p = 1.5$ and $\gamma_{\mathrm{i}} = 2^4$. For the $L_1$ cost, $\gamma_{\mathrm{f}} = 2^{16}$ and for the $L_2^2$ cost, $\gamma_{\mathrm{f}} = 2^{19}$.

In contrast, the $\varepsilon$-scaling approach of Schmitzer (2019) and Feydy (2020) maintains reparamatrized dual variables $\widetilde{\boldsymbol{z}} := \boldsymbol{z}/\gamma$ as the temperature decays. Rewriting (2) in terms of $\widetilde{\boldsymbol{z}}$ reveals that $\varepsilon$-scaling amounts to predicting $\boldsymbol{z}^*(\gamma^{(t+1)}) \approx (\gamma^{(t+1)}/\gamma^{(t)})\boldsymbol{z}^*(\gamma^{(t)})$, i.e., simply scaling the dual variables instead of modelling the trajectory of $\boldsymbol{z}^*$ with a Taylor approximation, which we argue is a better approach.

In Fig. 2, we present an empirical study with varying step sizes $\Delta_\gamma = (q-1)\gamma$ by ablating $q$, where the advantage of (12) over the $\varepsilon$-scaling warm-start of Schmitzer (2019) is demonstrated. On the left, MDOT warm-start initializes each dual problem closer to the solution than the $\varepsilon$-scaling approach. The quality of initial guesses increase markedly with decreasing temperature (left); at high temperatures dual problems are initialized very close to the solution with the gradient norm just a small multiple of the target $\varepsilon_{\mathrm{d}}$. In contrast, the $\varepsilon$-scaling warm-start stays relatively fixed. For the same decay rate $q$, this translates to about $10\times$ gains in convergence speed or precision (mid-right). The performance gap widens for slow temperature decay (lower $q$), as MDOT benefits from reduced Taylor approximation errors given smaller step sizes $\Delta_\gamma$.

### 4.2.2 KL Projection Stopping Criteria

Our assignment $\varepsilon_{\mathrm{d}} \propto \gamma^{-p}$ for some $p \geq 1$ in L4 of MDOT departs from the conventional wisdom of choosing $\varepsilon_{\mathrm{d}} \propto \gamma^{-1}$ (Altschuler et al., 2017; Dvurechensky et al., 2018; Lin et al., 2019). Here, we provide justification for this departure. Consider first the fixed-temperature problem (3) for simplicity. Building on the results of Cominetti & Martín (1994), Weed (2018) showed in his Prop. 4 and Thm. 5 that there is both a uniform bound $\langle P^*(\gamma) - P^*, C\rangle \leq \log n/\gamma$ (slow rate), and a fast asymptotic rate $O(\exp(-\gamma K))$ which takes over for large enough $\gamma$, where the constant $K > 0$ is problem-dependent. Taking these as a starting point, the following remark generalizes the third statement of Prop. 4.2.

**Remark 4.4.** *For any constant $p \in [1, \infty)$ and OT problem given by $(\boldsymbol{r}, \boldsymbol{c}, C)$, there exists a $\gamma_0 > 0$ such that for any $\gamma \geq \gamma_0$, we have $\langle P^*(\gamma) - P^*, C\rangle \leq H_{\min}(\boldsymbol{r}, \boldsymbol{c})/\gamma^p$.*

Figure 3: Ablation of stopping criterion parameter $p$ (Sec. 4.2.2) for the SK algorithm (**left**) and MDOT with Sinkhorn and PNCG as KL projectors (**right**). The SK algorithm (**left**) is called by running MDOT (Alg. 1) with $\gamma_i = \gamma_f$, where higher precision is achieved by increasing $\gamma_f$. Results show the median over 36 random problems from the upsampled MNIST dataset ($n = 4096$) with the $L_1$ cost.

That is, below some temperature $\gamma_0^{-1}$, a stronger bound $H_{\min}(\boldsymbol{r}, \boldsymbol{c})\gamma^{-p}$ for some $p > 1$ replaces the uniform bound $H_{\min}(\boldsymbol{r}, \boldsymbol{c})\gamma^{-1}$. Thus, the SK algorithm (see Alg. 4 in the Appx.) can be tuned (via the $p$ parameter in Alg. 1) to enjoy a rate substantially better than $O(n^2 \log n/\varepsilon^2)$ given by Dvurechensky et al. (2018).

**Proposition 4.5.** *Sinkhorn iteration, as instantiated by calling Alg. 1 (L6) with $p \in [1, \infty)$ and a sufficiently large $\gamma_i = \gamma_f = \sqrt[p]{5H_{\min}(\boldsymbol{r}, \boldsymbol{c})/2\varepsilon}$, returns a plan $P \in \mathcal{U}(\boldsymbol{r}, \boldsymbol{c})$ satisfying $\langle P - P^*, C \rangle \leq \varepsilon + \widetilde{O}(\varepsilon^2)$ in at most*

$$O\left(n^2 H_{\min}(\boldsymbol{r}, \boldsymbol{c})^{1/p} \Big/ \varepsilon^{\frac{p+1}{p}}\right) \text{ arithmetic operations.} \tag{13}$$

This result is consistent with the empirical findings of Jambulapati et al. (2019), who noted "The [tuned] Sinkhorn algorithm converged at rates much faster than the predicted $\varepsilon^{-2}$ rate on all experiments, outperforming all other methods, which we believe merits further investigation." We believe Prop. 4.5 sheds some light on this phenomenon, and further present an ablation of $p$ in Fig. 3 for SK and MDOT algorithms.

For the SK algorithm, Fig. 3 (left) verifies the insight derived from Prop. 4.5. The choice $p = 1$ is better at low precision, but the trend gradually shifts in favor of higher $p$ with (sufficiently) higher $\gamma_f$. That is, for sufficiently low temperature $\gamma^{-1}$, it is advantageous to reduce the gradient norm error tolerance, from $H_{\min}/\gamma$ to $H_{\min}/\gamma^p$ for $p > 1$. In contrast, MDOT is more robust to the $p$ parameter in the high precision regime (right). Moreover, the use of PNCG projections (Alg. 2) for KL projections in MDOT (L6 of Alg. 1) provides a speedup of $2 - 3\times$ over Sinkhorn projections. PNCG is introduced and discussed next.

### 4.3 Preconditioned Non-linear Conjugate Gradients for KL Projections

SK converges more slowly at low temperatures (Kosowsky & Yuille, 1994). For a faster alternative, we develop Alg. 2 based on non-linear CG (NCG) methods (Fletcher & Reeves, 1964; Nocedal & Wright, 2006), which we now briefly review. Given an objective $g$, NCG takes descent directions $\boldsymbol{p}^{(0)} = -\nabla g(\boldsymbol{z}^{(0)})$ and $\boldsymbol{p}^{(k)} \leftarrow -\nabla g^{(k)} + \beta^{(k)}\boldsymbol{p}^{(k-1)}$, and iterates $\boldsymbol{z}^{(k+1)} \leftarrow \boldsymbol{z}^{(k)} + \alpha^{(k)}\boldsymbol{p}^{(k)}$, where $\alpha^{(k)}$ is the step size. Optimal $\alpha^{(k)}$ has a closed-form solution for quadratics, but for general non-linear objectives, line search is necessary to find suitable step sizes $\alpha^{(k)}$. Many formulas for computing $\beta^{(k)}$ exist; for quadratic objectives, they are equivalent and guarantee convergence in at most $n'$ iterations, where $n' \leq n$ is the number of distinct eigenvalues of $\nabla^2 g$. Further, the objective decreases faster if eigenvalues are tightly clustered (Stiefel, 1958; Kaniel, 1966; Nocedal & Wright, 2006). For example, the Hestenes-Stiefel formula sets (Nocedal & Wright, 2006):

$$\beta^{(k)} = \frac{\langle \nabla g^{(k)} - \nabla g^{(k-1)}, \nabla g^{(k)} \rangle}{\langle \nabla g^{(k)} - \nabla g^{(k-1)}, \boldsymbol{p}^{(k-1)} \rangle}. \tag{14}$$

A practical way to further improve the convergence rate of CG methods is via *preconditioning.* By making a change of variables $\boldsymbol{z} = M^{-1/2}\hat{\boldsymbol{z}}$ given some symmetric positive-definite matrix $M$, one reduces the condition number of the problem or tightens the clustering of eigenvalues for improved convergence (ideally, $M^{-1} \approx \nabla^2 g^{-1}$). We refer the reader to Hager & Zhang (2006b) for further details on CG methods.

For the EOT problem, recall the 1st and 2nd order derivatives of the dual objective $g$ in (3) at $\boldsymbol{z} = (\boldsymbol{u}, \boldsymbol{v})$:

$$\nabla g = \big(\boldsymbol{r}(P) - \boldsymbol{r}, \ \ \boldsymbol{c}(P) - \boldsymbol{c}\big), \quad \nabla^2 g = \begin{pmatrix} \mathbf{D}(\boldsymbol{r}(P)) & P \\ P^\top & \mathbf{D}(\boldsymbol{c}(P)) \end{pmatrix}. \tag{15}$$

A typical choice of a preconditioner $M$, known to be effective for diagonally-dominant matrices (Golub & Van Loan, 2013), is the diagonal approximation of the Hessian, which yields the following descent direction:

$$\tilde{\boldsymbol{s}} = -\mathbf{D}\big(\mathbf{diag}(\nabla^2 g)\big)^{-1}\nabla g = \left(\frac{\boldsymbol{r}}{\boldsymbol{r}(P)}, \frac{\boldsymbol{c}}{\boldsymbol{c}(P)}\right) - \mathbf{1}_{2n}. \tag{16}$$

Observe, however, that if at any point in the optimization $\boldsymbol{r}(P)$ or $\boldsymbol{c}(P)$ has infinitesimal entries, numerical instabilities may occur when evaluating $\tilde{\boldsymbol{s}}$. We propose using the *Sinkhorn direction*, $\boldsymbol{s}$, in place of $\tilde{\boldsymbol{s}}$:

$$\boldsymbol{s} = \left(\log\frac{\boldsymbol{r}}{\boldsymbol{r}(P)}, \ \ \log\frac{\boldsymbol{c}}{\boldsymbol{c}(P)}\right). \tag{17}$$

This has the benefit that it can be evaluated via numerically stable LogSumExp reductions, e.g., see lines 2-3 of Alg. 2. The Sinkhorn direction can be understood as the result of an alternative diagonal preconditioner, namely $M = \mathbf{D}\left(-\nabla g/\boldsymbol{s}\right)$, since $\boldsymbol{s} = -M^{-1}\nabla g$. Furthermore, for any sub-optimal $(\boldsymbol{u}, \boldsymbol{v})$, we have

$$-\langle \boldsymbol{s}, \nabla g\rangle = D_{\mathrm{KL}}\big(\boldsymbol{r}(P)|\boldsymbol{r}\big) + D_{\mathrm{KL}}\big(\boldsymbol{r}|\boldsymbol{r}(P)\big) + D_{\mathrm{KL}}\big(\boldsymbol{c}(P)|\boldsymbol{c}\big) + D_{\mathrm{KL}}\big(\boldsymbol{c}|\boldsymbol{c}(P)\big) > 0,$$

and therefore $\boldsymbol{s}$ is also a descent direction. Empirically we find that this Sinkhorn preconditioner results in improved numerical stability. Finally, note that near the solution (for $\boldsymbol{r} \approx \boldsymbol{r}(P)$ and $\boldsymbol{c} \approx \boldsymbol{c}(P)$) we have

$$\boldsymbol{s} = \left(\log\frac{\boldsymbol{r}}{\boldsymbol{r}(P)}, \log\frac{\boldsymbol{c}}{\boldsymbol{c}(P)}\right) \approx \left(\frac{\boldsymbol{r}}{\boldsymbol{r}(P)}, \frac{\boldsymbol{c}}{\boldsymbol{c}(P)}\right) - \mathbf{1}_{2n} = \tilde{\boldsymbol{s}}, \tag{18}$$

where we have used $\log x \approx x - 1$ for $x \approx 1$. Therefore, near the solution, the Sinkhorn direction $\boldsymbol{s}$ approaches the direction $\tilde{\boldsymbol{s}}$ obtained using the common preconditioner from the diagonal of the Hessian.

Plugging the preconditioner $M = \mathbf{D}\left(-\nabla g/\boldsymbol{s}\right)$ into the preconditioned Hestenes-Stiefel formula (Al-Baali & Fletcher, 1996), we take $\beta^{(k)}$ in L7 of Alg. 2:

$$\beta^{(k)} = \frac{\langle \nabla g^{(k)} - \nabla g^{(k-1)}, -\boldsymbol{s}^{(k)}\rangle}{\langle \nabla g^{(k)} - \nabla g^{(k-1)}, \boldsymbol{p}^{(k-1)}\rangle}, \tag{19}$$

where $\boldsymbol{s}^{(k)}$ is the Sinkhorn direction as in (17), $\beta^{(1)} = 0$, $\boldsymbol{p}^{(0)} = \mathbf{0}_{2n}$. Observe that $-\boldsymbol{s}^{(k)}$ above simply replaces a $\nabla g^{(k)}$ term in the numerator of (14).

We defer details of the line search in L11 of Alg. 2 to Appx. B, but note that by design, the proposed line search only carries out the same form of LogSumExp reductions as the log-domain stabilized SK algorithm (Alg. 4 in the Appx A), so that its output is reused when evaluating the Sinkhorn direction $\boldsymbol{s}$ in (17) at the next iteration (see L11 of Alg. 2). This also allows for a fair comparison of the two algorithms' performance.

---

**Algorithm 2** PNCG$(\boldsymbol{z}, \gamma, C, \boldsymbol{r}, \boldsymbol{c}, \varepsilon_{\mathrm{d}})$

1: $(\boldsymbol{u}, \boldsymbol{v}) \leftarrow \boldsymbol{z}, \boldsymbol{p} \leftarrow \mathbf{0}_{2n}, \beta \leftarrow 0$
2: $\log\boldsymbol{r}(P) \leftarrow \boldsymbol{u} + \mathrm{LSE}_r(\mathbf{1}_n\boldsymbol{v}^\top - \gamma C)$
3: $\log\boldsymbol{c}(P) \leftarrow \boldsymbol{v} + \mathrm{LSE}_c(\boldsymbol{u}\mathbf{1}_n^\top - \gamma C)$
4: $\nabla g \leftarrow \big(\boldsymbol{r}(P) - \boldsymbol{r}, \boldsymbol{c}(P) - \boldsymbol{c}\big)$
5: **while** $\|\nabla g\|_1 > \varepsilon_{\mathrm{d}}$ **do**
6:     $\boldsymbol{s} \leftarrow \big(\log\boldsymbol{r} - \log\boldsymbol{r}(P), \log\boldsymbol{c} - \log\boldsymbol{c}(P)\big)$
7:     $\boldsymbol{p} \leftarrow \boldsymbol{s} + \beta\boldsymbol{p}$        ▷ See (19)
8:     **if** $\langle \boldsymbol{p}, \nabla g\rangle \geq 0$ **then**
9:         $\boldsymbol{p} \leftarrow \boldsymbol{s}$ ▷ Reset CG if not a descent dir.
10:     **end if**
11:     $\alpha, \log\boldsymbol{r}(P), \log\boldsymbol{c}(P) \leftarrow \mathrm{LineSearch}(\boldsymbol{p}, \boldsymbol{u}, \boldsymbol{v})$
12:     $(\boldsymbol{u}, \boldsymbol{v}) \leftarrow (\boldsymbol{u}, \boldsymbol{v}) + \alpha\boldsymbol{p}$
13:     $\nabla g \leftarrow \big(\boldsymbol{r}(P) - \boldsymbol{r}, \boldsymbol{c}(P) - \boldsymbol{c}\big)$
14: **end while**
15: Output $\boldsymbol{z} \leftarrow (\boldsymbol{u}, \boldsymbol{v})$

---

Indeed, Fig. 4 plots the runtime of the two algorithms in terms of LogSumExp evaluations; PNCG outshines SK empirically, especially at lower temperatures (higher $\gamma$). Further, to see whether the added numerical stability of the newly proposed Sinkhorn preconditioner comes at a performance trade-off, we implement an alternative stabilization scheme for the diagonal Hessian preconditioner. In particular, for this alternative we use $\boldsymbol{s}$ only if the vector $(\boldsymbol{r}/\boldsymbol{r}(P), \boldsymbol{c}/\boldsymbol{c}(P))$ has any entries outside the range $[0.01, 100]$ and otherwise assign $\tilde{\boldsymbol{s}}$ given by (16) in L6 of Alg. 2. The results shown in Fig. 4 suggest that, on the contrary, the Sinkhorn preconditioner also provides a performance benefit over the diagonal Hessian in addition to numerical stability.

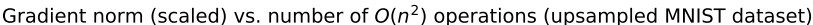

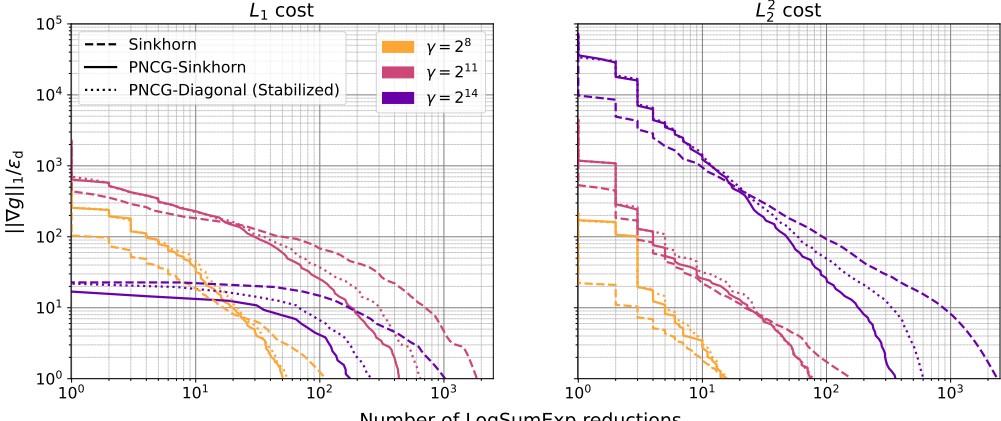

Figure 4: Comparison of KL projection algorithms (used in L7 of Alg. 1) over the upsampled MNIST dataset with $L_1$ (**left**) and $L_2^2$ (**right**) costs. Algorithms evaluated are the SK algorithm, the newly proposed PNCG given in Alg. 2, and a variant (see text). In all 36 problems, $n = 4096, p = 1.5$ and $q = 2$. Each curve shows the convergence behavior, at one specific temperature $1/\gamma$, in terms of the median number of LogSumExp reductions (x-axis) until gradient norm (y-axis) reaches below target dual gradient norm $\varepsilon_{\mathrm{d}}(\gamma)$.

## 5 EXPERIMENTS

In this section, we first detail the MNIST experimental setup in Figs. 2-4. Then, we describe an additional color transfer task we use for benchmarking. Next, performance evaluations in terms of precision vs. wall-clock time are discussed given the results over 4 sets of problems shown in Fig. 5. In Appx. H, we add 20 more problem sets from the DOTmark benchmark of Schrieber et al. (2017) showing similar results both in terms of wall-clock time and operation counts. Lastly, the dependence on problem size $n$ is investigated in Fig. 6. All experiments were performed on an NVIDIA GeForce RTX 2080 Ti GPU with 64-bit precision. Exact costs $\langle P^*, C \rangle$ are evaluated using the implementation of the CPU-based algorithm of Bonneel et al. (2011) from the Python Optimal Transport (POT) library (Flamary et al., 2021), which was run on an Intel Xeon Silver 4110 (2.10GHz) CPU.

### 5.1 Experimental Setup

**Upsampled MNIST.** In line with prior work (Cuturi, 2013; Altschuler et al., 2017; Lin et al., 2022; Tang et al., 2024), we first consider the MNIST dataset, where each pixel represents an event and each image a probability distribution. Unlike prior work, we form higher dimensional problems by upsampling the original $28 \times 28$ images to be $64 \times 64$ (with bilinear interpolation) so that $n = 4096$. Cost matrices $C$ are constructed by measuring the $L_1$ or squared $L_2$ distances between pixel locations on a 2D grid, and dividing all entries by the maximum distance value so that all entries of $C$ lie in $[0, 1]$. The probability of each pixel is proportional to its intensity value; marginals $\boldsymbol{r}, \boldsymbol{c}$ are obtained by flattening the pixel intensity matrices and subsequent $L_1$ normalization. To select $m$ random problems, we sample $2m$ images from the dataset without replacement, and compute the OT distances between the first and second halves of the samples. Our selection of $n = 4096$ is driven by the objective of conducting a large number of tests per configuration to ensure statistically significant results, rather than by any inherent limitations of the algorithm. In fact, our MDOT code supports the use of on-the-fly CUDA kernels to evaluate entries of the cost matrix on the go using the PyKeOps package (Charlier et al., 2021). In this case, MDOT leaves an $O(n)$ memory footprint (with both Sinkhorn and PNCG projections) rather than $O(n^2)$; it has been verified to scale to much larger problems ($n \approx 100,000$).

**Color Transfer.** For the color transfer problem, each image is viewed as a point cloud in RGB space (pixel locations carry no importance). Cost matrices $C$ are constructed by measuring the $L_1$ or $L_2^2$ distances between pixels in RGB space and dividing all entries by the maximum distance. Marginals $\boldsymbol{r}, \boldsymbol{c}$ are taken to be uniform over $\Delta_n$. With the help of GPT-4, we prompt DALL-E 2 to generate 20 vibrant and colorful images with intricate details or patterns. To match the dimensionality of the upsampled MNIST problem set, we downsample the original $1024 \times 1024$ images to $64 \times 64$ so that $n = 4096$. Once again, cost matrix entries are normalized to lie in $[0, 1]$.

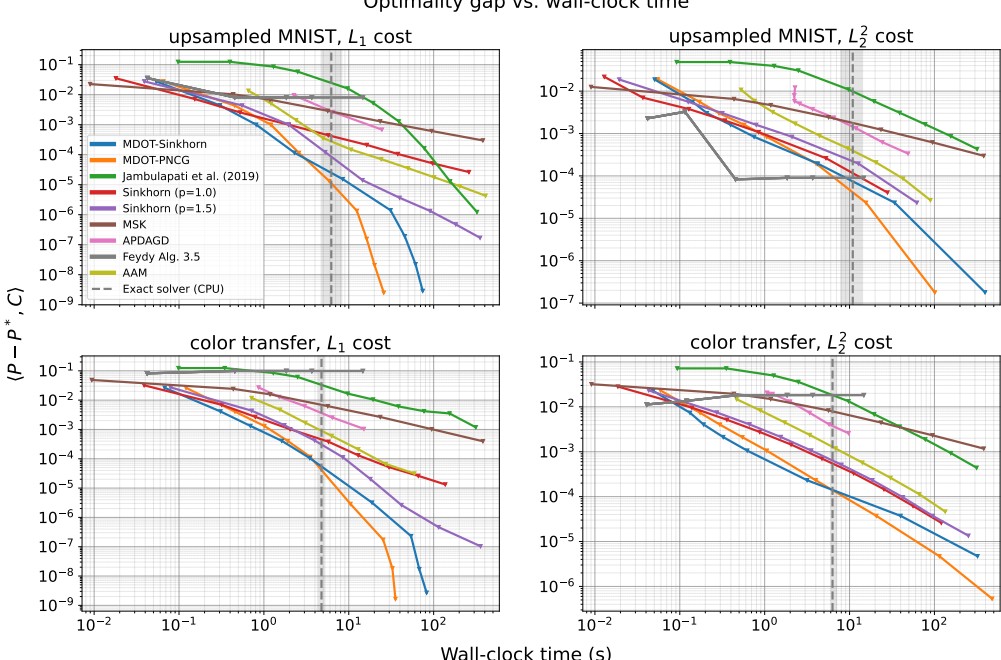

Figure 5: Wall-clock time vs. error benchmarking over the upsampled MNIST (top) and color transfer (bottom) problems using $L_1$ (left) and $L_2^2$ (right) distances as cost functions. Each marker shows the median time to converge (over 18 random problems) for each algorithm at a given hyperparameter setting, which controls the precision level, and the error $\langle P - P^*, C \rangle$ after rounding the output of the algorithm onto $\mathcal{U}(\boldsymbol{r}, \boldsymbol{c})$ – with the exception of Alg. 3.5 of Feydy (2020); see text. Vertical dashed lines show the median time taken by the CPU-based exact solver with 80% confidence intervals.

## 5.2 Wall-clock Time Comparisons With Prior Work

In Fig. 5, we present wall-clock time benchmarking of MDOT (with both Sinkhorn and PNCG projections) against existing GPU-parallel algorithms on the upsampled MNIST and color transfer problems. All benchmark methods were implemented in PyTorch and run on the GPU. For MDOT, we use $q = 2^{1/3}, p = 1.5$ and $\gamma_i = 2^4$ in all experiments. For the closely related Mirror Sinkhorn (MSK) algorithm of Ballu & Berthet (2023) the variable step size schedule prescribed by their Thm. 3.3 is used in our implementation. For Alg. 3.5 of Feydy (2020), we decay temperature at a rate $q = 0.7^{-1}$, which interpolates their *fast* ($q=0.5^{-1}$) and *safe* ($q=0.9^{-1}$) settings. For AAM (Guminov et al., 2021), Mirror Prox Sherman Optimized (Jambulapati et al., 2019) and APDAGD (Dvurechensky et al., 2018), each implementation closely follows an open-source NumPy implementation. Our PyTorch implementation was verified to produce identical results to the publicly available NumPy code. We additionally attempted comparison with APDAMD (Lin et al., 2019) and PDASMD (Luo et al., 2023), but observed extremely long convergence times for $n = 4096$ and omitted the results. For further details on the implementation of benchmark methods, see Appx. F.

While MDOT optimizes (3) to satisfy a convergence criterion following each temperature decrease, Alg. 3.5 of Feydy (2020) performs *a single* (symmetrized) Sinkhorn update instead, i.e., it does not minimize the sequence of dual objectives sufficiently despite taking increasingly large gradient steps in the dual space (cf. 5a-5b). This causes an accumulation of projection errors and results in the algorithm hitting a precision wall. Their *debiasing* option for estimating the OT distance via *Sinkhorn divergences* (introduced by Ramdas et al. (2017)) fares slightly better and is used here to comprise a stronger baseline, albeit this approach does not find a member of $\mathcal{U}(\boldsymbol{r}, \boldsymbol{c})$, which may be a strict requirement in some applications. MSK also runs a single row/column scaling update after a temperature decrease, but takes increasingly smaller steps and maintains a running average of transport plans to ensure convergence. It performs well at low precision, but shrinking step sizes slow it down, so that it exhibits $O(n^2 \varepsilon^{-2})$ convergence behavior. Sinkhorn iteration (log-domain stabilized, see Alg. 4 in the Appx. A) benefits substantially from setting $p = 1.5$ rather than $p = 1$ at sufficiently low temperatures for $L_1$ costs (see also Sec. 4.2.2). APDAGD underperforms SK with $p = 1$ and AAM performs similarly to it. Mirror Prox Sherman Optimized of Jambulapati et al. (2019) overtakes

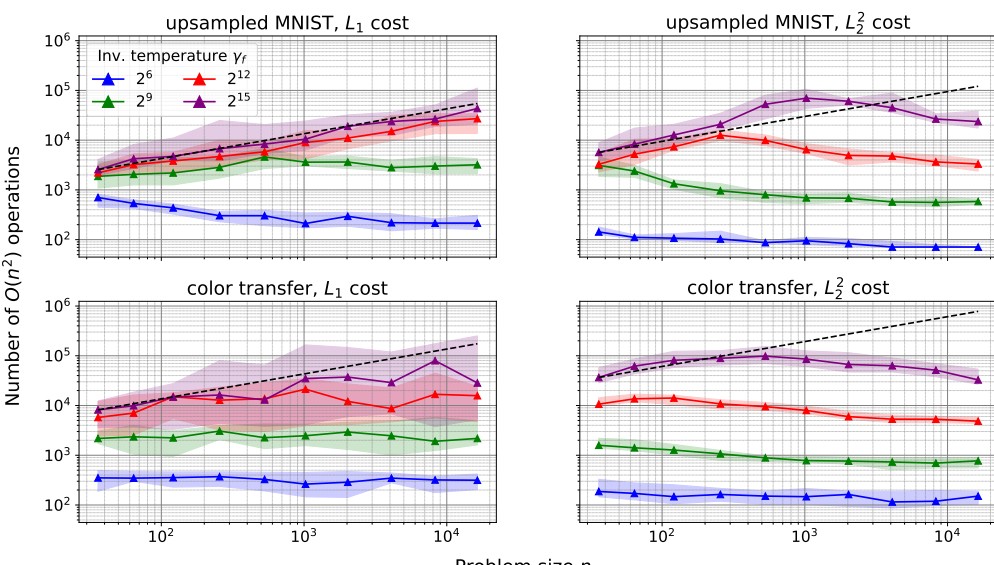

Figure 6: Problem size dependence of MDOT-PNCG convergence over the upsampled MNIST (top) and color transfer (bottom) problems using $L_1$ (left) and $L_2^2$ (right) distance cost functions with $\gamma_{\mathrm{f}} \in \{2^6, 2^9, 2^{12}, 2^{15}\}$. Each marker displays the median over 20 problems and shaded areas show 75% confidence intervals. Dashed lines show $f(n) = an^{5/2}$ for visual comparison, with $a$ selected to intersect the purple curve at the lowest $n$.

SK ($p = 1.0$) in one case only (top-left) in the high precision range. Meanwhile, MDOT-Sinkhorn enjoys faster convergence than the more competitive SK ($p = 1.5$) owing to warm-started temperature annealing, especially in the high precision range (near the solution). MDOT-PNCG is the quickest to converge in all cases. The performance gap with its close second, MDOT-Sinkhorn, grows with higher precision.

### 5.3 Empirical Dependence on Problem Size of MDOT-PNCG

Our last set of experiments investigates the practical dependence of MDOT-PNCG on the problem size $n$. Over the same 4 problem sets as Fig. 5, we change $n$ from 36 to 16, 384 by up- or down-sampling images. The $n$ values are selected to be approximately equally spaced on a logarithmic scale. In Fig. 6, we plot the behavior of MDOT-PNCG for a range of final temperature values $\gamma_{\mathrm{f}} \in \{2^6, 2^9, 2^{12}, 2^{15}\}$. At medium precision (green and blue), we find that the algorithm behaves no worse than $O(n^2)$ in practice as implied by the flatness of the curves. As seen visually, with higher precision (roughly 5-decimals) with $\gamma_{\mathrm{f}} \in \{2^{12}, 2^{15}\}$, the proposed GPU-parallel algorithm behaves roughly as $O(n^{5/2})$ at worst and even better for some of the problems in practice (see the reference line in Fig. 6). These should be compared to the $\widetilde{O}(n^{5/2})$ theoretical and $\widetilde{O}(n^3)$ practical complexity of CPU-based exact solvers (Pele & Werman, 2009; Lee & Sidford, 2014). Indeed, Figure 5 suggests that the CPU-based solver offers a strictly better precision-time trade-off beyond an optimality gap of around $10^{-4}$–$10^{-5}$ for this value of $n = 4096$ and this particular CPU-GPU setup; however, in Table 1 of the Appendix, we show that the trade-off skews in favor of MDOT-PNCG when $n$ is increased.

## 6 CONCLUSION

In this work, we first presented a general procedure, MDOT, for computing OT distances with high precision and described its relation to a well-known temperature annealing strategy ($\varepsilon$-scaling). MDOT employs a novel warm-starting of the sequence of EOT dual problems encountered in temperature annealing, which was empirically shown to be highly effective compared to existing approaches. In addition, a specialized non-linear CG algorithm was developed as an alternative to Sinkhorn iteration and was shown to be more effective at low temperatures (under weak regularization). Over 24 different problem sets, the combined MDOT-PNCG algorithm outperforms aggressively tuned Sinkhorn iteration and many other recent baselines in terms of convergence of the primal suboptimality gap measured in wall-clock time. The algorithm was also shown to behave well with respect to the problem size. Interesting directions for future research include the theoretical convergence behavior of PNCG, bounds on the gradient norm of our warm-started initialization,

better warm-starting methods, and adaptive dual problem stopping criteria. The development of faster KL projection algorithms and adaptive temperature decay schedules are also of interest; see also our recent work in this direction (Kemertas et al., 2025).

**Acknowledgements**

Amir-massoud Farahmand acknowledges the support of the Natural Sciences and Engineering Research Council of Canada (NSERC) through the Discovery Grant program (2021-03701). Mete Kemertas acknowledges the support of NSERC through the CGS-D scholarship. Resources used in preparing this research were provided, in part, by the Province of Ontario, the Government of Canada through CIFAR, and companies sponsoring the Vector Institute.

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

## Appendix

The Appendix is organized as follows:

## A   Proofs

Here, we provide proofs for the theoretical results in the main text; Lemma 4.1 in Appx. A.1, Proposition 4.2 in Appx. A.2 and Proposition 4.5 in Appx. A.3.

### A.1   Proof of Lemma 4.1

**Lemma 4.1.** *Given any initial $\boldsymbol{u}, \boldsymbol{v} \in \mathbb{R}^n$, $\gamma \geq 0$ and $\Delta_\gamma > 0$, we have*

$$P(\boldsymbol{u}^*, \boldsymbol{v}^*; \gamma + \Delta_\gamma) = \underset{P \in \mathcal{U}(\boldsymbol{r}, \boldsymbol{c})}{\arg\min} D_{\mathrm{KL}}\Big(P | P(\boldsymbol{u}, \boldsymbol{v}; \gamma) \odot \exp\{-\Delta_\gamma C\}\Big), \tag{9}$$

*where $\boldsymbol{u}^*, \boldsymbol{v}^* \in \arg\min g(\gamma + \Delta_\gamma, \boldsymbol{r}, \boldsymbol{c})$.*

*Proof.* First, we emphasize the following observation, from which Lemma 4.1 follows immediately using $P(\boldsymbol{u}, \boldsymbol{v}; \gamma) \odot \exp\{-\Delta_\gamma C\} = P(\boldsymbol{u}, \boldsymbol{v}; \gamma + \Delta_\gamma)$.

> **Observation A.1.** *Given any initial $\boldsymbol{u}, \boldsymbol{v} \in \mathbb{R}^n$ and $P(\boldsymbol{u}, \boldsymbol{v}; \gamma)$ defined as in (2), we have*
>
> $$P^*(\gamma) = \underset{P \in \mathcal{U}(\boldsymbol{r}, \boldsymbol{c})}{\arg\min} D_{\mathrm{KL}}\Big(P | P(\boldsymbol{u}, \boldsymbol{v}; \gamma)\Big) = P(\boldsymbol{u}^*, \boldsymbol{v}^*; \gamma), \;\; \text{where } \boldsymbol{u}^*, \boldsymbol{v}^* \in \arg\min g(\gamma, \boldsymbol{r}, \boldsymbol{c}). \tag{20}$$

In words, (i) minimizing the objective $g$ in (3) given any initial $\boldsymbol{u}, \boldsymbol{v} \in \mathbb{R}^n$ amounts to a KL projection of $P(\boldsymbol{u}, \boldsymbol{v}; \gamma)$ onto $\mathcal{U}(\boldsymbol{r}, \boldsymbol{c})$, and (ii) the set of matrices $P(\boldsymbol{u}, \boldsymbol{v}; \gamma) = \exp\{\boldsymbol{u}\mathbf{1}^\top + \mathbf{1}\boldsymbol{v}^\top - \gamma C\}$ all have the same KL projection in $\mathcal{U}(\boldsymbol{r}, \boldsymbol{c})$. The intuition here is that since the elements $u_i, v_j$ of $\boldsymbol{u}, \boldsymbol{v}$ simply rescale the $i^{\text{th}}$ row and $j^{\text{th}}$ column of $P(\boldsymbol{u}, \boldsymbol{v}; \gamma)$, and the unique projection onto $\mathcal{U}(\boldsymbol{r}, \boldsymbol{c})$ is the optimal scaling (with $\boldsymbol{r}(P) = \boldsymbol{r}$ and $\boldsymbol{c}(P) = \boldsymbol{c}$), the specific initial scaling of rows and columns in the conditioning matrix $P(\boldsymbol{u}, \boldsymbol{v}; \gamma)$ is irrelevant.

We now start the formal proof by deriving the relationship

$$P(\boldsymbol{u}^*, \boldsymbol{v}^*; \gamma) = \underset{P \in \mathcal{U}(\boldsymbol{r}, \boldsymbol{c})}{\arg\min} \big[ D_{\mathrm{KL}}\big(P | P(\boldsymbol{u}, \boldsymbol{v}; \gamma)\big) + K \big], \tag{21}$$

given any initial $\boldsymbol{u}, \boldsymbol{v} \in \mathbb{R}^n$ and $\boldsymbol{u}^*, \boldsymbol{v}^* \in \arg\min_{\boldsymbol{u}, \boldsymbol{v} \in \mathbb{R}^n} g(\boldsymbol{u}, \boldsymbol{v}; \gamma)$. Here we have introduced a constant $K$ which, of course, does not effect the arg min. We will choose $K$ to simplify the derivation below. Note that $K$ can depend on the given quantities, such as $\boldsymbol{u}$, $\boldsymbol{v}$, and so on, but not on the unknown $P$. Given the

definition of $D_{\mathrm{KL}}$ in Sec. 2 for un-normalized $P \in \mathbb{R}_{>0}^{n \times n}$, observe that

$$
\begin{aligned}
D_{\mathrm{KL}}\big(P | P(\boldsymbol{u}, \boldsymbol{v}; \gamma)\big) &= \langle P, \log P - \log P(\boldsymbol{u}, \boldsymbol{v}; \gamma)\rangle + \langle P(\boldsymbol{u}, \boldsymbol{v}; \gamma) - P, \mathbf{1}\rangle \\
&= \langle P, \log P - \boldsymbol{u}\mathbf{1}^\top - \mathbf{1}\boldsymbol{v}^\top + \gamma C\rangle + \langle P(\boldsymbol{u}, \boldsymbol{v}; \gamma) - P, \mathbf{1}\rangle \\
&= \langle P, -\mathbf{1} + \gamma C + \log P\rangle - \langle \boldsymbol{u}, \boldsymbol{r}(P)\rangle - \langle \boldsymbol{v}, \boldsymbol{c}(P)\rangle + \underbrace{\|P(\boldsymbol{u}, \boldsymbol{v}; \gamma)\|_1}_{\text{constant in } P}.
\end{aligned}
$$

As shown below, it is convenient to use $K = \langle \boldsymbol{u}, \boldsymbol{r}\rangle + \langle \boldsymbol{v}, \boldsymbol{c}\rangle - \|P(\boldsymbol{u}, \boldsymbol{v}; \gamma)\|_1$, for which we find

$$
D_{\mathrm{KL}}\big(P | P(\boldsymbol{u}, \boldsymbol{v}; \gamma)\big) + K = \langle P, -\mathbf{1} + \gamma C + \log P\rangle - \langle \boldsymbol{u}, \boldsymbol{r}(P) - \boldsymbol{r}\rangle - \langle \boldsymbol{v}, \boldsymbol{c}(P) - \boldsymbol{c}\rangle. \tag{22}
$$

We represent the equality constraints on $P$, that is, $\boldsymbol{r}(P) = \boldsymbol{r}$ and $\boldsymbol{c}(P) = \boldsymbol{c}$, using Lagrange multipliers $\boldsymbol{\alpha}$ and $\boldsymbol{\beta}$ to form the Lagrangian

$$
\begin{aligned}
\mathcal{L}(P, \boldsymbol{\alpha}, \boldsymbol{\beta}) &= \big[D_{\mathrm{KL}}\big(P | P(\boldsymbol{u}, \boldsymbol{v}; \gamma)\big) + K\big] - \langle \boldsymbol{\alpha}, \boldsymbol{r}(P) - \boldsymbol{r}\rangle - \langle \boldsymbol{\beta}, \boldsymbol{c}(P) - \boldsymbol{c}\rangle \\
&= \langle P, -\mathbf{1} + \gamma C + \log P\rangle - \langle \boldsymbol{u} + \boldsymbol{\alpha}, \boldsymbol{r}(P) - \boldsymbol{r}\rangle - \langle \boldsymbol{v} + \boldsymbol{\beta}, \boldsymbol{c}(P) - \boldsymbol{c}\rangle, \tag{23}
\end{aligned}
$$

where we have used $D_{\mathrm{KL}} + K$ as in (22). Recall that the Lagrange dual function is given by (Boyd & Vandenberghe, 2004):

$$
\inf_{P \in \mathbb{R}_{>0}^{n \times n}} \mathcal{L}(P, \boldsymbol{\alpha}, \boldsymbol{\beta}),
$$

where $\mathbb{R}_{>0}^{n \times n}$ is the domain of the KL divergence objective in (21). If strong duality holds, we have

$$
\sup_{\boldsymbol{\alpha}, \boldsymbol{\beta} \in \mathbb{R}^n} \inf_{P \in \mathbb{R}_{>0}^{n \times n}} \mathcal{L}(P, \boldsymbol{\alpha}, \boldsymbol{\beta}) = \min_{P \in \mathcal{U}(\boldsymbol{r}, \boldsymbol{c})} \big[D_{\mathrm{KL}}\big(P | P(\boldsymbol{u}, \boldsymbol{v}; \gamma)\big) + K\big]. \tag{24}
$$

In this case, we conclude that strong duality holds since Slater's condition is satisfied, i.e., there exists a strictly feasible $P$; namely, the independence coupling $\boldsymbol{r}\boldsymbol{c}^\top$. See Ch. 5.2.3 of Boyd & Vandenberghe (2004).

Now, to find the point-wise minimum of $\mathcal{L}(P, \boldsymbol{\alpha}, \boldsymbol{\beta})$ with respect to $P$, we require:

$$
\frac{\partial \mathcal{L}}{\partial P_{ij}} = \gamma C_{ij} + \log P_{ij} - u_i - \alpha_i - v_j - \beta_j = 0.
$$

Therefore the minimizer $P^*$ must have the form

$$
P_{ij}^* = P(\boldsymbol{u} + \boldsymbol{\alpha}, \boldsymbol{v} + \boldsymbol{\beta}; \gamma) = \exp(\alpha_i + u_i + \beta_j + v_j - \gamma C_{ij}). \tag{25}
$$

for some $\boldsymbol{\alpha}$ and $\boldsymbol{\beta}$. To eliminate the variable $P$ from $\inf_{P \in \mathbb{R}_{>0}^{n \times n}} \mathcal{L}(P, \boldsymbol{\alpha}, \boldsymbol{\beta})$, we plug the form above into the Lagrangian. The first term on the right hand side of (23) is then

$$
\begin{aligned}
\langle P^*, -\mathbf{1} + \gamma C + \log P^*\rangle &= \langle P^*, -\mathbf{1} + \gamma C\rangle + \langle P^*, (\boldsymbol{u} + \boldsymbol{\alpha})\mathbf{1}^T + \mathbf{1}(\boldsymbol{v} + \boldsymbol{\beta})^T - \gamma C\rangle, \\
&= -\langle P^*, \mathbf{1}\rangle + \langle \boldsymbol{u} + \boldsymbol{\alpha}, \boldsymbol{r}(P^*)\rangle + \langle \boldsymbol{v} + \boldsymbol{\beta}, \boldsymbol{c}(P^*)\rangle.
\end{aligned}
$$

Using this in (23), we find

$$
\begin{aligned}
\mathcal{L}(P^*, \boldsymbol{\alpha}, \boldsymbol{\beta}) &= -\langle P^*, \mathbf{1}\rangle + \langle \boldsymbol{u} + \boldsymbol{\alpha}, \boldsymbol{r}\rangle + \langle \boldsymbol{v} + \boldsymbol{\beta}, \boldsymbol{c}\rangle, \\
&= -\sum_{ij} \exp\big((u_i + \alpha_i) + (v_j + \beta_j) - \gamma C_{ij}\big) + \langle \boldsymbol{u} + \boldsymbol{\alpha}, \boldsymbol{r}\rangle + \langle \boldsymbol{v} + \boldsymbol{\beta}, \boldsymbol{c}\rangle. \tag{26}
\end{aligned}
$$

where $\boldsymbol{\alpha}$ and $\boldsymbol{\beta}$ are the only unknowns. Without loss of generality we can change variables in both $P^* = P(\boldsymbol{u} + \boldsymbol{\alpha}, \boldsymbol{v} + \boldsymbol{\beta}; \gamma)$ and $\mathcal{L}(P^*, \boldsymbol{\alpha}, \boldsymbol{\beta})$ to $\boldsymbol{\alpha}' = \boldsymbol{\alpha} + \boldsymbol{u}$ and $\boldsymbol{\beta}' = \boldsymbol{\beta} + \boldsymbol{u}$, in which case the Lagrangian is equal to

$$
\begin{aligned}
\mathcal{L}(P(\boldsymbol{\alpha}', \boldsymbol{\beta}'; \gamma), \boldsymbol{\alpha}', \boldsymbol{\beta}') &= -\sum_{ij} \exp(\alpha_i' + \beta_j' - \gamma C_{ij}) + \langle \boldsymbol{\alpha}', \boldsymbol{r}\rangle + \langle \boldsymbol{\beta}', \boldsymbol{c}\rangle \\
&= -g(\boldsymbol{\alpha}', \boldsymbol{\beta}'; \gamma), \tag{27}
\end{aligned}
$$

where $g(\boldsymbol{\alpha}', \boldsymbol{\beta}'; \gamma)$ is the dual objective for the EOT problem in (3). By strong duality and (24) we conclude that the solution is given by $P^* = P(\boldsymbol{\alpha}^*, \boldsymbol{\beta}^*; \gamma)$, where $\boldsymbol{\alpha}^*, \boldsymbol{\beta}^*$ satisfy:

$$
\begin{aligned}
\boldsymbol{\alpha}^*, \boldsymbol{\beta}^* &\in \underset{\boldsymbol{\alpha}, \boldsymbol{\beta} \in \mathbb{R}^n}{\arg\max} \, \mathcal{L}(P^*(\boldsymbol{\alpha}, \boldsymbol{\beta}), \boldsymbol{\alpha}, \boldsymbol{\beta}) \\
&= \underset{\boldsymbol{\alpha}, \boldsymbol{\beta} \in \mathbb{R}^n}{\arg\min} \, g(\boldsymbol{\alpha}, \boldsymbol{\beta}; \gamma).
\end{aligned}
$$

That is, the solution is identical to the optimal dual solution in (3), and therefore Observation A.1 holds.

Lemma 4.1 follows immediately from Observation A.1 using $P(\boldsymbol{u}, \boldsymbol{v}; \gamma) \odot \exp\{-\Delta_\gamma C\} = P(\boldsymbol{u}, \boldsymbol{v}; \gamma + \Delta_\gamma)$. $\blacksquare$

### A.2  Proof of Proposition 4.2

**Proposition 4.2.** *Let $\gamma^{(0)} = 0$ and $\gamma^{(t+1)} = \gamma^{(t)} + \Delta_\gamma^{(t)}$, which together imply $\gamma^{(t+1)} = \sum_{t'=0}^{t} \Delta_\gamma^{(t')}$. Suppose $P^{(0)} \in \mathbb{R}_{>0}^{n \times n}$ is rank-1 and $P^{(t)}$ are computed via (8) for $t \geq 0$. The following are true.*

*1. $P^{(t+1)} = P^*(\gamma^{(t+1)})$, i.e., the solution of the EOT problem (1) at $\gamma^{(t+1)}$.*

*2. Given any $\boldsymbol{u}, \boldsymbol{v} \in \mathbb{R}^n$, $P^{(t+1)} = \arg\min_{P \in \mathcal{U}(\boldsymbol{r}, \boldsymbol{c})} D_{\mathrm{KL}}(P | P(\boldsymbol{u}, \boldsymbol{v}; \gamma^{(t+1)}))$.*

*3. $\langle P^{(t)} - P^*, C \rangle \leq H_{\min}(\boldsymbol{r}, \boldsymbol{c}) / \gamma^{(t)}$, where $P^* \in \arg\min_{P \in \mathcal{U}(\boldsymbol{r}, \boldsymbol{c})} \langle P, C \rangle$ and $H_{\min}(\boldsymbol{r}, \boldsymbol{c}) := \min\big(H(\boldsymbol{r}), H(\boldsymbol{c})\big)$.*

*4. $\langle P^{(t)} - P^{(t+1)}, C \rangle = \frac{1}{\Delta_\gamma^{(t)}}\Big(D_{\mathrm{KL}}\big(P^{(t)} | P^{(t+1)}\big) + D_{\mathrm{KL}}\big(P^{(t+1)} | P^{(t)}\big)\Big)$ for all $t \geq 0$.*

*Proof.* We prove each of the four statements in order.

#### A.2.1  Proof of the 1st statement.

Here, we first provide a step by step derivation for the following expression for $P^{(t+1)}$ for $t \geq 0$:

$$
P^{(t+1)} = \underset{P \in \mathcal{U}(\boldsymbol{r}, \boldsymbol{c})}{\arg\min} \, D_{\mathrm{KL}}\Big(P | P\big(\boldsymbol{u}^*(\gamma^{(t)}), \boldsymbol{v}^*(\gamma^{(t)}); \gamma^{(t+1)}\big)\Big) = P^*(\gamma^{(t+1)}),
$$

where $\gamma^{(0)} = 0$ by construction, and $P^*(\gamma) = P(\boldsymbol{u}^*(\gamma), \boldsymbol{v}^*(\gamma); \gamma)$ for $\boldsymbol{u}^*(\gamma), \boldsymbol{v}^*(\gamma) \in \arg\min_{\boldsymbol{u}, \boldsymbol{v} \in \mathbb{R}^n} g(\gamma)$, i.e., solutions of the EOT primal (1) and dual (3) at $\gamma$. Starting with a rank-1 positive matrix $P^{(0)} = \tilde{\boldsymbol{r}} \tilde{\boldsymbol{c}}^\top = P(\log \tilde{\boldsymbol{r}}, \log \tilde{\boldsymbol{c}}; 0)$ given vectors $\tilde{\boldsymbol{r}}, \tilde{\boldsymbol{c}}$ with positive entries, we obtain using MD updates (8) and Lemma 4.1:

$$
\begin{aligned}
P^{(1)} &= \underset{P \in \mathcal{U}(\boldsymbol{r}, \boldsymbol{c})}{\arg\min} \, D_{\mathrm{KL}}\Big(P | P\big(\log \tilde{\boldsymbol{r}}, \log \tilde{\boldsymbol{c}}; \Delta_\gamma^{(0)}\big)\Big) \\
&= \underset{P \in \mathcal{U}(\boldsymbol{r}, \boldsymbol{c})}{\arg\min} \, D_{\mathrm{KL}}\Big(P | P\big(\log \tilde{\boldsymbol{r}}, \log \tilde{\boldsymbol{c}}; \gamma^{(1)}\big)\Big) \quad \text{(Since } \gamma^{(0)} = 0 \text{ and } \gamma^{(t+1)} = \gamma^{(t)} + \Delta_\gamma^{(t)} \text{ by construction.)} \\
&= P^*(\boldsymbol{u}^*(\gamma^{(1)}), \boldsymbol{v}^*(\gamma^{(1)}); \gamma^{(1)}) \quad\quad\quad\quad\quad\quad\quad\quad\quad\quad\quad\quad\quad\quad\quad\quad\text{(By Observation A.1.)} \\
&= P^*(\gamma^{(1)}). \quad\quad\quad\quad\quad\quad\quad\quad\quad\quad\quad\quad\quad\quad\quad\quad\quad\quad\quad\quad\quad\quad\quad\text{(By definition.)}
\end{aligned}
$$

Repeatedly using the same and continuing the iteration:

$$
\begin{aligned}
P^{(2)} &= \underset{P \in \mathcal{U}(\boldsymbol{r}, \boldsymbol{c})}{\arg\min} \, D_{\mathrm{KL}}\Big(P | P\big(\boldsymbol{u}^*(\gamma^{(1)}), \boldsymbol{v}^*(\gamma^{(1)}); \gamma^{(2)}\big)\Big) = P^*(\gamma^{(2)}), \\
P^{(3)} &= \underset{P \in \mathcal{U}(\boldsymbol{r}, \boldsymbol{c})}{\arg\min} \, D_{\mathrm{KL}}\Big(P | P\big(\boldsymbol{u}^*(\gamma^{(2)}), \boldsymbol{v}^*(\gamma^{(2)}); \gamma^{(3)}\big)\Big) = P^*(\gamma^{(3)}), \\
&\qquad\qquad\qquad\qquad\qquad\qquad \vdots \\
P^{(t+1)} &= \underset{P \in \mathcal{U}(\boldsymbol{r}, \boldsymbol{c})}{\arg\min} \, D_{\mathrm{KL}}\Big(P | P\big(\boldsymbol{u}^*(\gamma^{(t)}), \boldsymbol{v}^*(\gamma^{(t)}); \gamma^{(t+1)}\big)\Big) = P^*(\gamma^{(t+1)}).
\end{aligned}
\tag{28}
$$

Hence, each iterate $P^{(t+1)} = P^*(\gamma^{(t+1)})$ for $t \geq 0$. $\blacksquare$

### A.2.2 Proof of the 2nd statement.

The result follows immediately by applying Observation A.1 to each iterate in (28) to replace $\boldsymbol{u}^*(\gamma), \boldsymbol{v}^*(\gamma)$ terms by arbitrary $\boldsymbol{u}, \boldsymbol{v} \in \mathbb{R}^n$. ∎

### A.2.3 Proof of the 3rd statement.

First, we write the following helper lemma.

**Lemma A.2** (A mirror descent bound for linear objectives). *Given a linear objective function $f(P) = \langle P, C \rangle$, an initial point $P^{(0)} \in \mathcal{F}$, an optimal point $P^*$ and any $T > 0$, a sequence $[P^{(t)}]_{t \in \mathbb{N}}$ obtained via (6) satisfies:*

$$f(P^{(T)}) - f(P^*) \leq \frac{D_h(P^*|P^{(0)})}{\sum_{t=0}^{T-1} \Delta^{(t)}}. \tag{29}$$

*Proof.* Recall the definition of $\hat{P}^{(t+1)}$ from mirror descent iterates in (5a):

$$\hat{P}^{(t+1)} = \nabla h^{-1}\left(\nabla h(P^{(t)}) - \Delta^{(t)} \nabla f(P^{(t)})\right)$$

For any $P \in \mathcal{D}$,

$$
\begin{aligned}
f(P^{(t+1)}) - f(P) &= \langle \nabla f(P^{(t)}), P^{(t+1)} - P \rangle && \text{(since } f \text{ is linear)} \\
&= \frac{1}{\Delta^{(t)}} \langle \nabla h(P^{(t)}) - \nabla h(\hat{P}^{t+1}), P^{(t+1)} - P \rangle && \text{(due to (5a))} \\
&\leq \frac{1}{\Delta^{(t)}} \langle \nabla h(P^{(t)}) - \nabla h(P^{(t+1)}), P^{(t+1)} - P \rangle && \text{(by Lemma 4.1 in Bubeck (2015))} \\
&= \frac{1}{\Delta^{(t)}} \left(D_h(P|P^{(t)}) - D_h(P|P^{(t+1)}) - D_h(P^{(t+1)}|P^{(t)})\right) && \text{(by Eq. 4.1 in Bubeck (2015))} \\
&\leq \frac{1}{\Delta^{(t)}} \left(D_h(P|P^{(t)}) - D_h(P|P^{(t+1)})\right), && \text{(since } D_h \geq 0)
\end{aligned}
$$

which implies

$$\Delta^{(t)}\left(f(P^{(t+1)}) - f(P)\right) \leq D_h(P|P^{(t)}) - D_h(P|P^{(t+1)}).$$

The above inequality proves monotonic improvement in each step $t$ once we take $P = P^{(t)}$. Letting $P = P^*$, taking a telescopic sum and dividing both sides by $\sum_{s=0}^{T-1} \Delta^{(s)}$ we arrive at:

$$
\begin{aligned}
\frac{\sum_{t=0}^{T-1} \Delta^{(t)}\left(f(P^{(t+1)}) - f(P^*)\right)}{\sum_{s=0}^{T-1} \Delta^{(s)}} &\leq \frac{D_h(P^*|P^{(0)}) - D_h(P^*|P^{(T)})}{\sum_{s=0}^{T-1} \Delta^{(s)}} \\
&\leq \frac{D_h(P^*|P^{(0)})}{\sum_{s=0}^{T-1} \Delta^{(s)}},
\end{aligned}
$$

which implies (29) since improvement is monotonic and the first term on the LHS is a convex combination of objective values. ∎

By Lemma A.2, for $P^{(0)} \in \mathcal{U}(\boldsymbol{r}, \boldsymbol{c})$ we have

$$\langle P^{(t)} - P^*, C \rangle \leq \frac{D_h(P^*|P^{(0)})}{\sum_{t'=0}^{t-1} \Delta_\gamma^{(t')}}.$$

Given $\gamma = \gamma^{(t)} = \sum_{t'=0}^{t-1} \Delta_\gamma^{(t')}$, it remains to show that $D_h(P^*|P^{(0)}) \leq H_{\min}(\boldsymbol{r}, \boldsymbol{c})$.

Recall that for the negative entropy $h(\boldsymbol{x}) = \sum_i x_i \log x_i$, we have $D_h(\boldsymbol{x}|\boldsymbol{y}) = D_{\text{KL}}(\boldsymbol{x}|\boldsymbol{y})$. Suppose we take $P^{(0)} = \boldsymbol{r}\boldsymbol{c}^\top$:

$$
\begin{aligned}
D_{\text{KL}}(P^*|P^{(0)}) &= \sum_{ij} P_{ij}^*(\log P_{ij}^* - \log r_i c_j) \\
&= \sum_{ij} P_{ij}^*(\log P_{ij}^* - \log r_i - \log c_j) \\
&= -H(P^*) - \sum_i \log r_i \sum_j P_{ij}^* - \sum_j \log c_j \sum_i P_{ij}^* \\
&= -H(P^*) - \sum_i r_i \log r_i - \sum_j c_j \log c_j \qquad \text{(since } P^* \in \mathcal{U}(\boldsymbol{r}, \boldsymbol{c})) \\
&= H(\boldsymbol{r}) + H(\boldsymbol{c}) - H(P^*) \\
&= \max(H(\boldsymbol{r}), H(\boldsymbol{c})) + \min(H(\boldsymbol{r}), H(\boldsymbol{c})) - H(P^*) \\
&\leq \min(H(\boldsymbol{r}), H(\boldsymbol{c})).
\end{aligned}
$$

The last inequality holds since $H(P) \geq H(\boldsymbol{r})$ and $H(P) \geq H(\boldsymbol{c})$ for any $P \in \mathcal{U}(\boldsymbol{r}, \boldsymbol{c})$ (Cover, 1999), which together imply $H(P) \geq \max(H(\boldsymbol{r}), H(\boldsymbol{c}))$. ∎

**Proof of the 4th statement.** First, note that given $h(P) = \sum_{ij} P_{ij} \log P_{ij}$, we have $\nabla h(P)_{ij} = 1 + \log P_{ij}$ and $\nabla h^{-1}(Q)_{ij} = \exp(Q_{ij} - 1)$. Then, given the definition of $\hat{P}^{(t+1)}$ from mirror descent iterates in (5a):

$$
\begin{aligned}
\hat{P}^{(t+1)} &= \nabla h^{-1}\left(\nabla h(P^{(t)}) - \Delta_\gamma^{(t)} \nabla f(P^{(t)})\right) \\
&= \exp(\log P^{(t)} - \Delta_\gamma^{(t)} C) \\
&= \exp\left(\boldsymbol{u}^*(\gamma^{(t)})\mathbf{1}^\top + \mathbf{1}\boldsymbol{v}^*(\gamma^{(t)})^\top - (\gamma^{(t)} + \Delta_\gamma^{(t)})C\right). \quad \text{(given } \boldsymbol{u}^*(\gamma^{(t)}), \boldsymbol{v}^*(\gamma^{(t)}) \in \arg\min g(\boldsymbol{u}, \boldsymbol{v}; \gamma^{(t)})) \\
&= \exp\left(\boldsymbol{u}^*(\gamma^{(t)})\mathbf{1}^\top + \mathbf{1}\boldsymbol{v}^*(\gamma^{(t)})^\top - \gamma^{(t+1)}C\right). \quad\quad\quad\quad\quad\quad\quad (30)
\end{aligned}
$$

In the third equality, we used the known closed-form expression (2) to expand $P^{(t)}$.

In the special case that the feasible set $\mathcal{F} = \mathcal{U}(\boldsymbol{r}, \boldsymbol{c})$,

$$
\begin{aligned}
&\langle P^{(t)}, C\rangle - \langle P^{(t+1)}, C\rangle \\
&= \langle \nabla f(P^{(t)}), P^{(t)} - P^{(t+1)}\rangle && \text{(since } f(P) = \langle P, C\rangle \text{ is linear, } \nabla_P f(P) = C.) \\
&= \frac{1}{\Delta_\gamma^{(t)}}\langle \nabla h(P^{(t)}) - \nabla h(\hat{P}^{t+1}), P^{(t)} - P^{(t+1)}\rangle && \text{(due to (5a))} \\
&= \frac{1}{\Delta_\gamma^{(t)}}\langle \nabla h(P^{(t)}) - \nabla h(P^{(t+1)}), P^{(t)} - P^{(t+1)}\rangle && \text{(see below)} \\
&= \frac{1}{\Delta_\gamma^{(t)}}\left(D_h(P^{(t)}|P^{(t+1)}) + D_h(P^{(t+1)}|P^{(t)})\right) && \text{(by definition of the Bregman divergence as in (4))} \\
&= \frac{1}{\Delta_\gamma^{(t)}}\left(D_{\text{KL}}(P^{(t)}|P^{(t+1)}) + D_{\text{KL}}(P^{(t+1)}|P^{(t)})\right).
\end{aligned}
$$

To see why the third equality holds, observe that $P_{ij}^{t+1} = \hat{P}_{ij}^{t+1}\exp\{\hat{u}_i^* + \hat{v}_j^*\}$ for some optimal update vectors $\hat{\boldsymbol{u}}^*, \hat{\boldsymbol{v}}^* \in \mathbb{R}^n$ given the closed-forms (2) and (30). Then, for any $P, P' \in \mathcal{U}(\boldsymbol{r}, \boldsymbol{c})$,

$$
\begin{aligned}
&\langle \nabla h(P^{(t+1)}), P - P'\rangle \\
&= \sum_{ij}(1 + \log \hat{P}_{ij}^{t+1} + \hat{u}_i^* + \hat{v}_j^*)(P_{ij} - P_{ij}') \\
&= \langle \nabla h(\hat{P}^{t+1}), P - P'\rangle + \sum_i \hat{u}_i^* \sum_j (P_{ij} - P_{ij}') + \sum_j \hat{v}_j^* \sum_i (P_{ij} - P_{ij}') \\
&= \langle \nabla h(\hat{P}^{t+1}), P - P'\rangle + \langle \hat{\boldsymbol{u}}^*, \boldsymbol{r} - \boldsymbol{r}\rangle + \langle \hat{\boldsymbol{v}}^*, \boldsymbol{c} - \boldsymbol{c}\rangle && \text{(since } P, P' \in \mathcal{U}(\boldsymbol{r}, \boldsymbol{c}) \text{ by construction)} \\
&= \langle \nabla h(\hat{P}^{t+1}), P - P'\rangle. &&\quad\quad\quad ∎
\end{aligned}
$$

**Algorithm 3** Round($P, \boldsymbol{r}, \boldsymbol{c}$) (Altschuler et al., 2017)

1: $X \leftarrow \boldsymbol{D}(\boldsymbol{x})$ with $\boldsymbol{x} = \boldsymbol{r}/\boldsymbol{r}(P) \wedge 1$
2: $F \leftarrow XP$
3: $Y \leftarrow \boldsymbol{D}(\boldsymbol{y})$ with $\boldsymbol{y} = \boldsymbol{c}/\boldsymbol{c}(F) \wedge 1$
4: $F' \leftarrow FY$
5: $\mathrm{err}_r \leftarrow \boldsymbol{r} - \boldsymbol{r}(F'), \mathrm{err}_c \leftarrow \boldsymbol{c} - \boldsymbol{c}(F')$
6: Output $G \leftarrow F' + \mathrm{err}_r \mathrm{err}_c^\top / \|\mathrm{err}_r\|_1$

**Algorithm 4** Sinkhorn($\boldsymbol{z}, \gamma, C, \boldsymbol{r}, \boldsymbol{c}, \varepsilon_\mathrm{d}$)

1: $(\boldsymbol{u}, \boldsymbol{v}) \leftarrow \boldsymbol{z}$
2: $\log \boldsymbol{r}(P) \leftarrow \boldsymbol{u} + \mathrm{LSE}_r(\boldsymbol{1}_n \boldsymbol{v}^\top - \gamma C)$
3: **while** $\|\nabla g\|_1 = \|\boldsymbol{r} - \boldsymbol{r}(P)\|_1 > \varepsilon_\mathrm{d}$ **do**
4:     $\boldsymbol{u} \leftarrow \boldsymbol{u} + \log \boldsymbol{r} - \log \boldsymbol{r}(P)$
5:     $\boldsymbol{v} \leftarrow \log \boldsymbol{c} - \mathrm{LSE}_c(\boldsymbol{u}\boldsymbol{1}_n^\top - \gamma C)$
6:     $\log \boldsymbol{r}(P) \leftarrow \boldsymbol{u} + \mathrm{LSE}_r(\boldsymbol{1}_n \boldsymbol{v}^\top - \gamma C)$
7: **end while**
8: Output $\boldsymbol{z} \leftarrow (\boldsymbol{u}, \boldsymbol{v})$

### A.3 Proof of Proposition 4.5

In the remainder of this section, the $L_1$ norm $\|P\|_1$ of a matrix denotes the $L_1$ norm of the vectorized form of the matrix, and not the $L_1$ matrix norm.

First we state the following lemma, which is a simple combination of Lemmas 6 and 8 by Weed (2018).

**Lemma A.3** (Entropy increase from mixing (Weed, 2018)). *Let $\boldsymbol{r}_1, \boldsymbol{r}_2, \boldsymbol{r}_3 \in \Delta_n$ and $\boldsymbol{r}_2 = (1 - \varepsilon)\boldsymbol{r}_1 + \varepsilon \boldsymbol{r}_3$, where $\varepsilon \in (0, 1]$. We have,*

$$H(\boldsymbol{r}_2) \leq (1 - \varepsilon)H(\boldsymbol{r}_1) + \varepsilon H(\boldsymbol{r}_3) + \varepsilon(1 - \log \varepsilon) < H(\boldsymbol{r}_1) + \varepsilon(1 + \log \frac{n}{\varepsilon}). \tag{31}$$

Next, we provide a simple proof for Remark 4.4

**Remark 4.4.** *For any constant $p \in [1, \infty)$ and OT problem given by $(\boldsymbol{r}, \boldsymbol{c}, C)$, there exists a $\gamma_0 > 0$ such that for any $\gamma \geq \gamma_0$, we have $\langle P^*(\gamma) - P^*, C \rangle \leq H_{\min}(\boldsymbol{r}, \boldsymbol{c})/\gamma^p$.*

*Proof.* Recall from Thm. 5 of Weed (2018) that the quantity $\langle P^*(\gamma) - P^*, C \rangle$ decays at an exponential rate with increasing $\gamma$ for sufficiently large $\gamma$. Since the exponential function $\exp\{-\gamma K\}$ decays more quickly than $\gamma^{-p}$ for any constant $K > 0$ and finite $p$, we conclude that there exists some constant $\gamma_0 > 0$ such that

$$\langle P^*(\gamma) - P^*, C \rangle \leq H_{\min}(\boldsymbol{r}, \boldsymbol{c})/\gamma^p \tag{32}$$

for all optimal transport problems given by $\boldsymbol{r}, \boldsymbol{c}, C$ provided that $\gamma \geq \gamma_0$. ∎

**Proposition 4.5.** *Sinkhorn iteration, as instantiated by calling Alg. 1 (L6) with $p \in [1, \infty)$ and a sufficiently large $\gamma_\mathrm{i} = \gamma_\mathrm{f} = \sqrt[p]{5H_{\min}(\boldsymbol{r}, \boldsymbol{c})/2\varepsilon}$, returns a plan $P \in \mathcal{U}(\boldsymbol{r}, \boldsymbol{c})$ satisfying $\langle P - P^*, C \rangle \leq \varepsilon + \widetilde{O}(\varepsilon^2)$ in at most*

$$O\left(n^2 H_{\min}(\boldsymbol{r}, \boldsymbol{c})^{1/p} \middle/ \varepsilon^{\frac{p+1}{p}}\right) \text{ arithmetic operations.} \tag{13}$$

*Proof.* Let $B \in \mathcal{U}(\boldsymbol{r}', \boldsymbol{c}')$ be the transport plan $P(\boldsymbol{u}, \boldsymbol{v}) = \exp\{\boldsymbol{u}\boldsymbol{1}^\top + \boldsymbol{1}\boldsymbol{v}^\top - \gamma_\mathrm{f} C\}$ after the termination of the main loop (before rounding in L13) of Alg. 1, which takes place after a single outer loop iteration in this setting, since $\gamma_\mathrm{i} = \gamma_\mathrm{f}$ by construction. Since $B$ is the output of Sinkhorn iteration (Alg. 4), it lies on the simplex, as do its row and column marginals (specifically, we have $\boldsymbol{c}' = \boldsymbol{c}(B) = \tilde{\boldsymbol{c}} \in \Delta_n$ from Alg. 4). Furthermore, $B$ is the unique optimizer of the EOT problem over $\mathcal{U}(\boldsymbol{r}', \boldsymbol{c}')$ due to Prop. 4.2 and the fact that it has the form $B_{ij} = \exp\{\boldsymbol{u}_i + \boldsymbol{v}_j - \gamma_\mathrm{f} C_{ij}\}$:

$$B = \operatorname*{arg\,min}_{P \in \mathcal{U}(\boldsymbol{r}', \boldsymbol{c}')} \langle P, C \rangle - \frac{1}{\gamma_\mathrm{f}} H(P). \tag{33}$$

Sinkhorn iteration returns a solution $\boldsymbol{u}, \boldsymbol{v}$ such that

$$\|\tilde{\boldsymbol{r}} - \boldsymbol{r}(B)\|_1 + \|\tilde{\boldsymbol{c}} - \boldsymbol{c}(B)\|_1 \leq \varepsilon'/2$$
$$\implies \|\nabla g\|_1 = \|\boldsymbol{r} - \boldsymbol{r}(B)\|_1 + \|\boldsymbol{c} - \boldsymbol{c}(B)\|_1$$
$$\leq \|\boldsymbol{r} - \tilde{\boldsymbol{r}}\|_1 + \|\tilde{\boldsymbol{r}} - \boldsymbol{r}(B)\|_1 + \|\boldsymbol{c} - \tilde{\boldsymbol{c}}\|_1 + \|\tilde{\boldsymbol{c}} - \boldsymbol{c}(B)\|_1 \qquad \text{(triangle inequality)}$$
$$\leq \|\boldsymbol{r} - \tilde{\boldsymbol{r}}\|_1 + \|\boldsymbol{c} - \tilde{\boldsymbol{c}}\|_1 + \varepsilon'/2.$$

Then, given mixing weights $\varepsilon'/4$ in L5 of Alg. 1:

$$\|\nabla g\|_1 \le \varepsilon'. \tag{34}$$

Now, we make the following definitions:

- $\widehat{B} = \text{Round}(B, \boldsymbol{r}, \boldsymbol{c})$, the rounding of $B$ onto $\mathcal{U}(\boldsymbol{r}, \boldsymbol{c})$ via Alg. 2 of Altschuler et al. (2017), returned by our Alg. 1,
- $B^* \in \arg\min_{P \in \mathcal{U}(\boldsymbol{r}', \boldsymbol{c}')} \langle P, C \rangle$, an optimal plan in the feasible set $\mathcal{U}(\boldsymbol{r}', \boldsymbol{c}')$,
- $P^* \in \arg\min_{P \in \mathcal{U}(\boldsymbol{r}, \boldsymbol{c})} \langle P, C \rangle$, an optimal plan in the feasible set $\mathcal{U}(\boldsymbol{r}, \boldsymbol{c})$.

We have that,

$$
\begin{aligned}
\langle \widehat{B} - P^*, C \rangle &= \langle \widehat{B} - B, C \rangle + \langle B - B^*, C \rangle + \langle B^* - P^*, C \rangle \\
&= \langle \widehat{B} - B, C - \tfrac{1}{2}\mathbf{1}_{n \times n} \rangle + \langle B - B^*, C \rangle + \langle B^* - P^*, C \rangle &&\text{(since } \widehat{B}, B \in \Delta_{n \times n}.) \\
&\le \tfrac{1}{2}\left\|\widehat{B} - B\right\|_1 + \langle B - B^*, C \rangle + \langle B^* - P^*, C \rangle &&\text{(Hölder's ineq., given } C_{ij} \in [0,1]\ \forall i,j \in [n]) \\
&\le \|\nabla g\|_1 + \langle B - B^*, C \rangle + \langle B^* - P^*, C \rangle &&\text{(by Lemma 7 of Altschuler et al. (2017))} \\
&\le \|\nabla g\|_1 + \frac{H_{\min}(\boldsymbol{r}', \boldsymbol{c}')}{\gamma_{\mathrm{f}}^p} + \langle B^* - P^*, C \rangle &&\text{(given (32-33), assuming } \gamma_{\mathrm{f}} \text{ sufficiently large)} \\
&\le \varepsilon' + \frac{H_{\min}(\boldsymbol{r}', \boldsymbol{c}')}{\gamma_{\mathrm{f}}^p} + \langle \widetilde{B} - P^*, C \rangle, \tag{35}
\end{aligned}
$$

where $\widetilde{B}$ is any transport plan in $\mathcal{U}(\boldsymbol{r}', \boldsymbol{c}')$. We take $\widetilde{B}$ to be the "shadow" of $P^*$ in the sense of Definition 3.1 of Eckstein & Nutz (2022), under *the discrete metric*. In other words, letting

$$\widetilde{B} = \arg\min_{P \in \mathcal{U}(\boldsymbol{r}', \boldsymbol{c}')} \|P - P^*\|_1,$$

and noting that the 1-Wasserstein distance under the discrete metric is equal to the total variation (TV) distance, the first equation in Lemma 3.2 of Eckstein & Nutz (2022) yields the equality $(*)$ below:

$$\frac{1}{2}\left\|\widetilde{B} - P^*\right\|_1 = \text{TV}(B, P^*) \overset{(*)}{=} \text{TV}(\boldsymbol{r}, \boldsymbol{r}') + \text{TV}(\boldsymbol{c}, \boldsymbol{c}') = \frac{1}{2}\|\nabla g\|_1. \tag{36}$$

Then, continuing from (35),

$$
\begin{aligned}
\langle \widehat{B} - P^*, C \rangle &\le \varepsilon' + \frac{H_{\min}(\boldsymbol{r}', \boldsymbol{c}')}{\gamma_{\mathrm{f}}^p} + \langle \widetilde{B} - P^*, C \rangle \\
&= \varepsilon' + \frac{H_{\min}(\boldsymbol{r}', \boldsymbol{c}')}{\gamma_{\mathrm{f}}^p} + \langle \widetilde{B} - P^*, C - \tfrac{1}{2}\mathbf{1}_{n \times n} \rangle &&\text{(since } \widetilde{B}, P^* \in \Delta_{n \times n}.) \\
&= \varepsilon' + \frac{H_{\min}(\boldsymbol{r}', \boldsymbol{c}')}{\gamma_{\mathrm{f}}^p} + \left\|\widetilde{B} - P^*\right\|_1 \left\|C - \tfrac{1}{2}\mathbf{1}_{n \times n}\right\|_\infty \\
&\le \frac{3}{2}\varepsilon' + \frac{H_{\min}(\boldsymbol{r}', \boldsymbol{c}')}{\gamma_{\mathrm{f}}^p} &&\text{(given (34-36))} \\
&\le \frac{3}{2}\varepsilon' + \frac{H_{\min}(\boldsymbol{r}, \boldsymbol{c})}{\gamma_{\mathrm{f}}^p} + \frac{\varepsilon'}{\gamma_{\mathrm{f}}^p}\left(1 + \log(n/\varepsilon')\right) &&\text{(by Lemma A.3)} \\
&= \frac{5 H_{\min}(\boldsymbol{r}, \boldsymbol{c})}{2\gamma_{\mathrm{f}}^p} + \widetilde{O}(\gamma_{\mathrm{f}}^{-2p}) &&\text{(since } \varepsilon' = \frac{H_{\min}(\boldsymbol{r}, \boldsymbol{c})}{\gamma_{\mathrm{f}}^p} \text{ in L4 of Alg. 1)} \\
&= \varepsilon + \widetilde{O}(\varepsilon^2). &&\text{(since } \gamma_{\mathrm{f}} = \left(5 H_{\min}(\boldsymbol{r}, \boldsymbol{c})/2\varepsilon\right)^{1/p} \text{ by construction)}
\end{aligned}
$$

The computational complexity of the algorithm follows simply from the same line of reasoning as Thm. 1 and Thm. 2 of Dvurechensky et al. (2018). In particular, they show that Sinkhorn iteration converges in $O(R/\varepsilon')$

steps, where $R = O(\gamma_f) = O(H_{\min}(\boldsymbol{r}, \boldsymbol{c})^{1/p} \varepsilon^{-1/p})$ in our case. The complexity result $O(n^2 H_{\min}(\boldsymbol{r}, \boldsymbol{c})^{1/p}/\varepsilon^{\frac{p+1}{p}})$ follows since $\varepsilon' = O(H_{\min}(\boldsymbol{r}, \boldsymbol{c})\gamma_f^{-p}) = O(\varepsilon^{-1})$, and each Sinkhorn step costs $O(n^2)$.

∎

## B   An Efficient Line Search Algorithm

In Section 4.3, we developed the PNCG algorithm, which required a line search procedure. Here, we develop the line search used in our implementation, following a short background on relevant aspects of line search in numerical optimization.

### B.1   Background: Line Search

Given a descent direction $\boldsymbol{p}^{(k)} \in \mathbb{R}^n$, i.e., a direction that satisfies $\langle \boldsymbol{p}^{(k)}, \nabla f(\boldsymbol{x}^{(k)}) \rangle \leq 0$, line search algorithms aim to find an appropriate step size $\alpha$, where $\boldsymbol{x}^{(k+1)} \leftarrow \boldsymbol{x}^{(k)} + \alpha \boldsymbol{p}^{(k)}$. Perhaps the most well-known of desirable properties that a step size $\alpha$ should satisfy at any given optimization step are the Wolfe conditions (Wolfe, 1969; 1971). Given $\phi(\alpha) := f(\boldsymbol{x}^{(k)} + \alpha \boldsymbol{p}^{(k)})$:

$$\frac{\phi(\alpha) - \phi(0)}{\alpha} \leq c_1 \phi'(0) \tag{37a}$$

$$\phi'(\alpha) \geq c_2 \phi'(0). \tag{37b}$$

where $0 < c_1 < c_2 < 1$ and (37a) and (37b) are known as the *sufficient decrease* and *curvature* conditions respectively (Nocedal & Wright, 2006). It is well-known that given step sizes satisfying the Wolfe conditions and descent directions $\boldsymbol{p}^{(k)}$ that are *not* nearly orthogonal to the steepest descent directions $-\nabla f(\boldsymbol{x}^{(k)})$, line search methods ensure convergence of the gradient norms to zero (Zoutendijk, 1966; Wolfe, 1969; 1971). Instead of satisfying (37), some algorithms or theoretical analyses consider *exact* line search, where $\alpha^* \in \arg\min_{\alpha \in \mathbb{R}} \phi(\alpha)$, which has a unique closed-form solution for quadratic objectives with a positive definite Hessian. However, a rule of thumb for general non-linear objectives is to not spend too much time finding $\alpha^*$ (Nocedal & Wright, 2006). Hager & Zhang (2006a) proposed *approximate* Wolfe conditions, derived by replacing the $\phi(\alpha)$ and $\phi(0)$ terms in (37a) with $q(\alpha)$ and $q(0)$, where $q$ is a quadratic interpolant of $\phi$ such that $q(0) = \phi(0)$, $q'(0) = \phi'(0)$ and $q'(\alpha) = \phi'(\alpha)$:

$$(2c_1 - 1)\phi'(0) \geq \phi'(\alpha) \geq c_2 \phi'(0). \tag{38}$$

A key advantage of replacing (37) by (38) stems from the fact that one only needs to evaluate $\phi'$ rather than both $\phi$ and $\phi'$ to check whether the conditions are satisfied, thereby halving the amount of computation necessary per iteration in cases where their evaluation has similar computational cost.

Bisection is a line search strategy with convergence guarantees when the objective is convex. One simply maintains a bracket $[\alpha_{\mathrm{lo}}, \alpha_{\mathrm{hi}}]$, where $\phi'(\alpha_{\mathrm{lo}}) < 0$ and $\phi'(\alpha_{\mathrm{hi}}) > 0$, and recursively considers their average and updates either endpoint of the bracket given the sign of $\phi'\big((\alpha_{\mathrm{hi}} + \alpha_{\mathrm{lo}})/2\big)$.

### B.2   PNCG Line Search

To perform line search in PNCG (Alg. 2), we adopt a hybrid strategy combining bisection and the secant method to find $\alpha_k$ satisfying approximate Wolfe conditions (38). Given $\alpha_{\mathrm{lo}}, \alpha_{\mathrm{hi}}$, the secant method computes the minimizer of a quadratic interpolant $\hat{q}$ that satisfies $\hat{q}'(\alpha_{\mathrm{lo}}) = \phi'(\alpha_{\mathrm{lo}})$ and $\hat{q}'(\alpha_{\mathrm{hi}}) = \phi'(\alpha_{\mathrm{hi}})$ as follows:

$$\alpha_{\mathrm{sec}} = \frac{\alpha_{\mathrm{lo}} \phi'(\alpha_{\mathrm{hi}}) - \alpha_{\mathrm{hi}} \phi'(\alpha_{\mathrm{lo}})}{\phi'(\alpha_{\mathrm{hi}}) - \phi'(\alpha_{\mathrm{lo}})}. \tag{39}$$

Thanks to the convexity of the objective $g$, by ensuring $\phi'(\alpha_{\mathrm{lo}}) < 0$ and $\phi'(\alpha_{\mathrm{hi}}) > 0$ with simple algorithmic checks, we can guarantee that $\alpha_{\mathrm{lo}} < \alpha_{\mathrm{sec}} < \alpha_{\mathrm{hi}}$. Thus, the updated bracket is guaranteed to be smaller once we replace either of $\alpha_{\mathrm{lo}}$ or $\alpha_{\mathrm{hi}}$ by $\alpha_{\mathrm{sec}}$ for the next bracket given the sign of $\phi'(\alpha_{\mathrm{sec}})$. If $\phi$ behaves like a quadratic inside the bracket, the secant method converges very quickly, but convergence can be arbitrarily slow otherwise. For this reason, we simply average the bisection estimate and $\alpha_{\mathrm{sec}}$ for a less aggressive but more reliable line search that still converges quickly, i.e., $\alpha_{\mathrm{hybrid}} = 0.5\alpha_{\mathrm{sec}} + 0.5(\alpha_{\mathrm{hi}} + \alpha_{\mathrm{lo}})/2$.

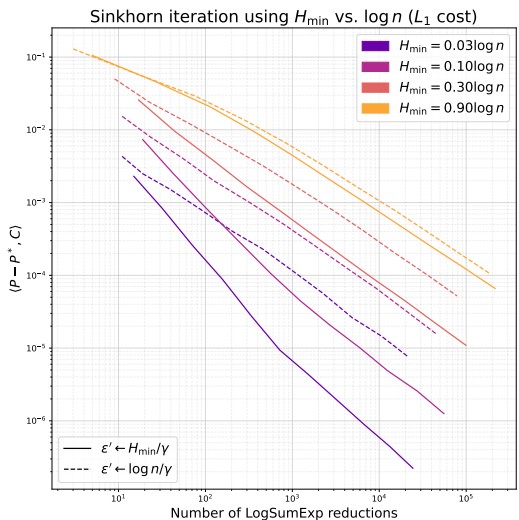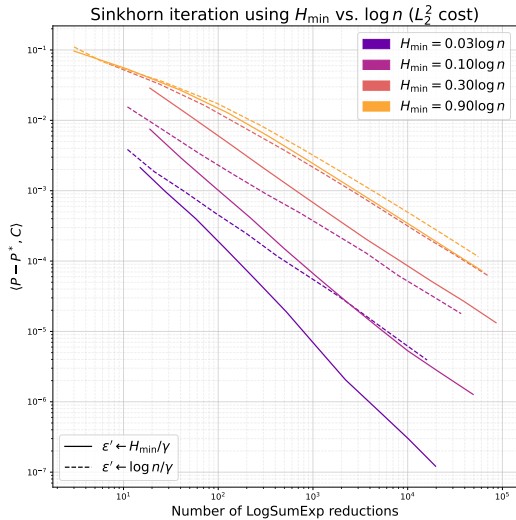

Figure 7: We control the problem parameter $H_{\min}(\boldsymbol{r}, \boldsymbol{c})$ over synthetically sampled OT problems $(\boldsymbol{r}, \boldsymbol{c}, C)$ and show that using the entropy-aware stopping criterion can yield substantial performance gains, with the gap between the two approaches growing proportionally to the gap between $H_{\min}(\boldsymbol{r}, \boldsymbol{c})$ and $\log n$. The experiments carried out use $\gamma \in \{2^4, 2^5, \cdots, 2^{14}\}$ to control precision. We sample 18 problems for each $\gamma$ value and plot the median along both axes. The results are consistent between $L_1$ (left) and $L_2^2$ (right) costs.

Evaluation of $\phi'$ has computational complexity $O(n^2)$ as does a single step of Sinkhorn's algorithm (given by the two LogSumExp reductions seen in L5-6 of Alg. 4):

$$\phi'(\alpha) = \langle \boldsymbol{p_u}, \boldsymbol{r}(P_\alpha) - \boldsymbol{r} \rangle + \langle \boldsymbol{p_v}, \boldsymbol{c}(P_\alpha) - \boldsymbol{c} \rangle, \tag{40}$$

where $(\boldsymbol{p_u}, \boldsymbol{p_v})$ is the descent direction. Since evaluating $\phi'$ requires the computation of $\boldsymbol{r}(P_\alpha)$ and $\boldsymbol{c}(P_\alpha)$ for the new matrix $P_\alpha := \exp\{(\boldsymbol{u} + \alpha \boldsymbol{p_u})\mathbf{1}^\top + \mathbf{1}(\boldsymbol{v} + \alpha \boldsymbol{p_v})^\top - \gamma C\}$, the last step of the line search readily carries out the LogSumExp reductions necessary for computing the Sinkhorn direction in the next step of PNCG (see L11 of Alg. 2). Observe also that at the next PNCG iteration, $\phi'(0)$ can also be computed in $O(n)$ time rather than $O(n^2)$ since $\boldsymbol{r}(P_0), \boldsymbol{c}(P_0)$ are already known from the last line search step of the previous PNCG iteration. With these important implementation details in place, we find that the average number of $\phi'$ evaluations necessary to find an $\alpha$ that satisfies (38) is typically between $1.5 - 2.5$ for the PNCG algorithm. While the approach outlined here is easy to implement (including as a batch process) and works well in practice, better line search methods may further benefit Alg. 2.

## C  Entropy-aware Stopping Criteria on the Dual Objective Gradient Norm

Here, we show the effect of choosing $H_{\min}(\boldsymbol{r}, \boldsymbol{c})$ over the weaker bound $\log n$ in L4 of Alg. 1, where the stopping criterion $\varepsilon'$ is selected. To control the problem setting $H_{\min}(\boldsymbol{r}, \boldsymbol{c})$, we construct synthetic problems by randomly sampling $\boldsymbol{r}$ from the simplex via a Dirichlet distribution constructed to meet a target entropy level $H(\boldsymbol{r})$ as a fraction of the maximum possible entropy $\log n$. The column marginal $\boldsymbol{c}$ is simply taken to be the uniform distribution $\mathbf{1}_n/n$, so that $H_{\min}(\boldsymbol{r}, \boldsymbol{c}) = H(\boldsymbol{r})$. Cost matrices are constructed by sampling $n$ points $\boldsymbol{x} \in \mathbb{R}^3$ from a multivariate normal distribution and assigning $C_{ij} = \|\boldsymbol{x}_i - \boldsymbol{x}_j\|_r^r$ for $r \in \{1, 2\}$, before entrywise division by $\max_{ij} C_{ij}$ to ensure $C_{ij} \in [0, 1]$.

Fig. 7 illustrates the effect of this choice by ranging $H_{\min}(\boldsymbol{r}, \boldsymbol{c})/\log n \in \{0.03, 0.1, 0.3, 0.9\}$. Towards the RHS of the plots, we observe an improvement in precision roughly proportional to $\log n / H_{\min}(\boldsymbol{r}, \boldsymbol{c})$ for the same number of operations, which agree with our complexity result $O(n^2 H_{\min}(\boldsymbol{r}, \boldsymbol{c})/\varepsilon^{-2})$ for $p = 1$ in (32) vs. the $O(n^2 \log n/\varepsilon^{-2})$ result by Dvurechensky et al. (2018).

## D  Variable vs. Fixed Smoothing of the Marginals in MDOT

As discussed in Sec. 4.2, MDOT smoothes the marginals $\boldsymbol{r}, \boldsymbol{c}$ (by mixing in the uniform distribution) with a weighting factor that tracks the temperature. Since MDOT anneals the temperature, this means that the smoothing weight is higher in earlier iterations of MDOT. In particular, the mixture weight gradually

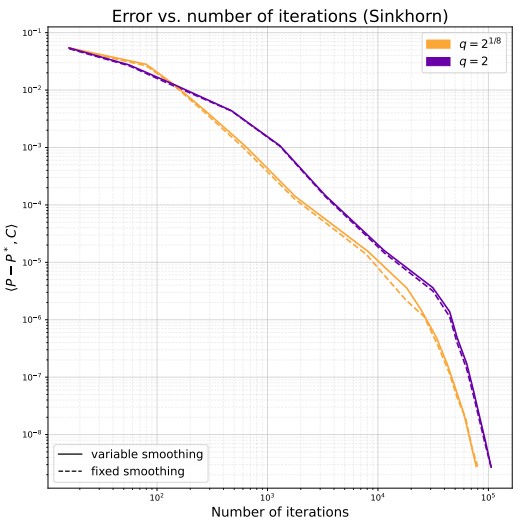 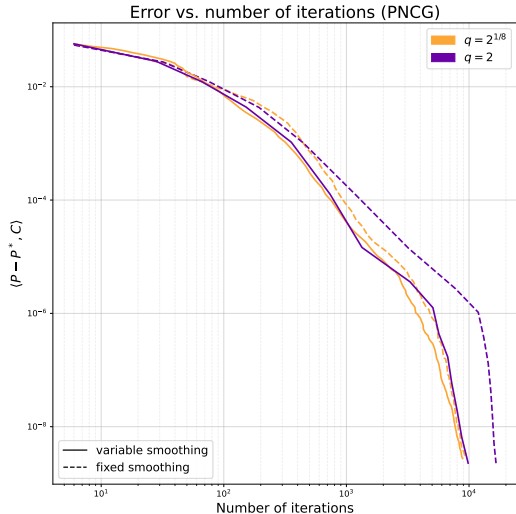

Figure 8: The variable smoothing scheme has almost no impact on the convergence behavior of MDOT-Sinkhorn **(left)** for both settings $q = 2$ (rapid temperature decay) and $q = 2^{1/8}$ (slow temperature decay). MDOT-PNCG **(right)** enjoys a speedup of nearly $2\times$ under rapid decay and a more modest speedup under slow decay. All curves show the median over 36 sample problems from the MNIST dataset ($L_1$ cost).

decays from $H_{\min}(\boldsymbol{r}, \boldsymbol{c})\big/4\gamma_{\mathrm{i}}^p$ to $H_{\min}(\boldsymbol{r}, \boldsymbol{c})\big/4\gamma_{\mathrm{f}}^p$ given input parameter $p \geq 1$. Here, we study the effect of this design choice as it influences the convergence of two KL projection algorithms used in L6 of Alg. 1: Sinkhorn iteration and the newly proposed PNCG algorithm (Alg. 2). The approach is benchmarked against a baseline that fixes the smoothing weight at $H_{\min}(\boldsymbol{r}, \boldsymbol{c})\big/4\gamma_{\mathrm{f}}^p$ all throughout instead. For these experiments, we fix $p = 1.5$ following our experimental setup in Sec. 5. Fig. 8 shows that while MDOT-Sinkhorn is largely unaffected by this design choice, MDOT-PNCG enjoys a notable speedup from variable smoothing. We thus conclude that the approach provides a performance benefit.

## E Comparison with CPU-based Solver for Higher $n$

In Figure 5, we displayed via a vertical line the time taken on a CPU by the network simplex solver from the Python Optimal Transport package of Flamary et al. (2021). Clearly, a faster CPU would benefit the network simplex and a faster GPU would benefit the MDOT family of algorithms, which makes them difficult to compare in a fair setup. However, given the known $\widetilde{O}(n^3)$ complexity of the network simplex and the $\widetilde{O}(n^2) - \widetilde{O}(n^{2.5})$ empirical dependence of MDOT-PNCG seen in Fig. 6, we suspect that the precision-speed trade-off posed by MDOT-PNCG improves with increasing $n$. In Table 1, we provide a comparison of the two algorithms, with several values of $\gamma_{\mathrm{f}}$ used for MDOT-PNCG and $n$ increased from $4,096$ to $16,384$. We observe the same trend across both $L_1$ and $L_2$ cost functions on the MNIST dataset; for a comparable level of relative error achieved by MDOT–PNCG (at a fixed $\gamma_{\mathrm{f}}$), the speedup over the network simplex solver is better for $n = 16,384$. We expect the trend to continue with higher $n$, and note that MDOT stands to benefit from ongoing rapid developments in GPU hardware innovation, as well as faster projection algorithms than PNCG; see also our concurrent work in this direction showing substantial speedups (Kemertas et al., 2025).

## F Details of Baseline Algorithm Implementations

Here, we provide details and sources on the implementation of various algorithms shown in Fig. 5. Our implementations of other algorithms will be open-sourced for transparency.

**Mirror Prox Sherman Optimized** (Jambulapati et al., 2019). For this algorithm, the source code is originated in the NumPy code at this repository. The owner of the repository notes that this NumPy implementation is based on a Julia implementation by the original authors, which was provided in a private exchange. The code used in this paper is a PyTorch adaptation of the NumPy code and has been verified to produce identical output as the NumPy version over multiple problems. The algorithm was called with *entropy factor* parameter set to the default 2.75 in all experiments. The number of iterations for the algorithm was varied from 2 to $2^{15}$ to achieve different levels of precision.

**APDAGD** (Dvurechensky et al., 2018). For APDAGD, a similar strategy was used, except with this code repository. A PyTorch version of the original NumPy code was written and verified to produce identical output. For different levels of precision, the $\varepsilon$ parameter of the algorithm was varied from $2^{-1}$ to $2^{-6}$. For smaller $\varepsilon$, non-convergence was observed.

**AAM** (Guminov et al., 2021). The implementation is based on NumPy code by the original authors at this repository. A PyTorch version was verified to produce identical output for GPU execution. The $\varepsilon$ parameter was varied from $2^{-1}$ to $2^{-10}$. For smaller $\varepsilon$, numerical errors were encountered.

**Feydy, Alg. 3.5** (Feydy, 2020). The implementation is based on the algorithm as presented in the original work. For different levels of precision, the number of total iterations was varied from 2 to $2^{12}$. Beyond the upper bound, numerical errors were observed. As it produced better estimates than the alternative, the algorithm was called with *debiasing* turned on; hence, the error $\langle P - P^*, C \rangle$ was instead measured in absolute value as $|\langle P - P^*, C \rangle|$ for this algorithm only. Scaling ratio was set to an intermediate 0.7, which is between the listed 0.5 (fast) and 0.9 (safe) settings.

**Sinkhorn** (Cuturi, 2013). A log-domain stabilized implementation was used. For different precision levels, $\gamma$ was varied from $2^5$ to $2^{14}$ for $L_1$ distance cost and to $2^{15}$ for $L_2^2$ distance cost. Stopping criteria were given by our formula in L4 of Alg. 1, and the results obtained by calling Alg. 1 with $\gamma_i = \gamma_f$, so that the algorithm terminates after a single KL projection via SK iteration.

**Mirror Sinkhorn (MSK)** (Ballu & Berthet, 2023). The implementation is based on the algorithm presented in the original paper. For different levels of precision, the number of total iterations was varied from $2^5$ to $2^{16}$.

Table 1: Comparison of the exact Network Simplex solver with MDOT–PNCG method on MNIST transport problems under $L_1$ and $L_2$ costs. Relative error is computed as $100 * \langle P - P^*, C \rangle / \langle P^*, C \rangle$.

| Cost Fn. | Algorithm | $\gamma_f$ | $n = 4,096$ | | | $n = 16,384$ | | |
|---|---|---|---|---|---|---|---|---|
| | | | **RelErr %** | **Time (s)** | **Speedup** | **RelErr %** | **Time (s)** | **Speedup** |
| | Net. Simplex | – | 0.0000 | 5.45 | 1.00 | 0 | 236.46 | 1.00 |
| | | $2^6$ | 16.556 | 0.18 | 30.55 | 22.580 | 1.72 | 137.24 |
| $L_1$ | MDOT–PNCG | $2^9$ | 0.167 | 2.28 | 2.39 | 0.585 | 26.94 | 8.78 |
| | | $2^{12}$ | 0.002 | 11.85 | 0.46 | 0.002 | 318.58 | 0.74 |
| | Net. Simplex | – | 0.000 | 11.74 | 1.00 | 0 | 728.40 | 1.00 |
| | | $2^9$ | 26.877 | 0.52 | 22.66 | 23.827 | 5.53 | 131.83 |
| $L_2$ | MDOT–PNCG | $2^{12}$ | 3.166 | 3.85 | 3.04 | 3.372 | 38.12 | 19.11 |
| | | $2^{15}$ | 0.044 | 39.38 | 0.30 | 0.303 | 294.12 | 2.48 |

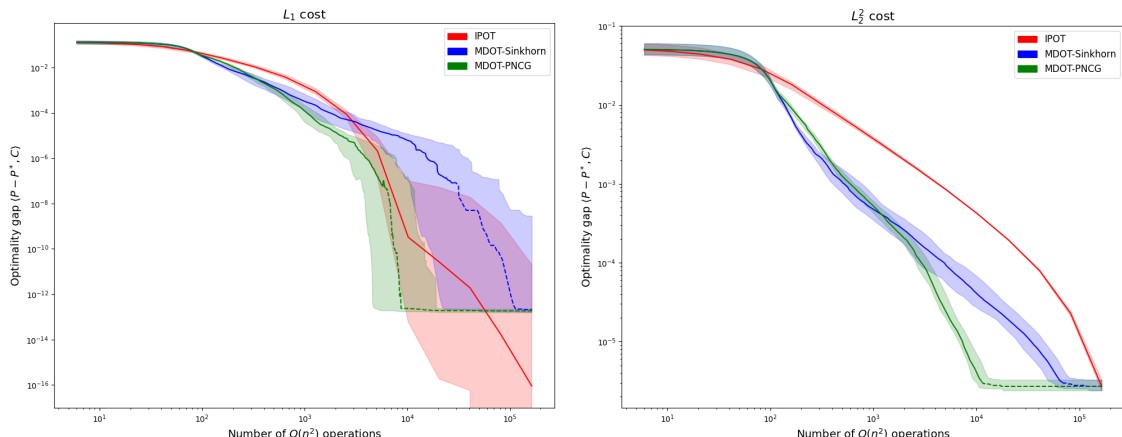

Figure 9: Optimality gap $\langle P - P^*, C \rangle$ of rounded (feasible) plans vs. number of $O(n^2)$ operations for IPOT (Xie et al., 2020) and MDOT (with Sinkhorn and PNCG for KL projections, using $p = 1.5, q = 2^{1/3}$ as in Section 5). Both methods are run up to a final $\gamma_{\rm f} = 2^{15}$ on the upsampled MNIST dataset ($n = 4096$) with $L_1$ (left) and $L_2^2$ (right) cost functions. MDOT reaches the target $\gamma_{\rm f}$ satisfying its stopping criterion $\|\nabla g(\gamma_{\rm f})\|_1 = O(\gamma^{-p})$ in fewer operations than IPOT takes until termination, so we continue the KL projection at $\gamma_{\rm f}$ until the same number of operations as IPOT is reached, without increasing $\gamma$ further (dashed lines).

# G Similarities and Differences with IPOT

In this section, we discuss the similarities and differences between MDOT and the Inexact Proximal point method for exact Optimal Transport (IPOT) algorithm of Xie et al. (2020) in more detail. In IPOT, authors propose to update (in our notation) the $\gamma$ parameter by fixed increments and typically choose $\Delta_\gamma^{(t)} = 1$ for all $t \geq 0$, while we took $\Delta_\gamma = (q-1)\gamma$ for a hyperparameter $q > 1$. Following each temperature update, they run a fixed number of $L$ row+column scaling (Sinkhorn) updates and recommend $L = 1$, whereas MDOT requires that the KL projection is continued until the dual objective gradient norm reaches below a threshold at each $\gamma$. A small number of $L$ Sinkhorn updates clearly does not amount to an "exact" KL projection onto the feasible set $\mathcal{U}(\boldsymbol{r}, \boldsymbol{c})$, but Xie et al. (2020) show that under some conditions there exists some finite $L$ that guarantees linear convergence of $\langle P(\boldsymbol{u}, \boldsymbol{v}; \gamma), C \rangle$ to the $\langle P^*, C \rangle$. On the other hand, **(i)** MDOT naturally adapts the number of optimization updates to the difficulty of the problem and the strictness of the stopping criterion at $\gamma$, and **(ii)** it stands to benefit from potentially faster convex optimization algorithms besides Sinkhorn iteration, e.g., our PNCG approach proposed in Sec. 4 or any other approach that may be developed in the future. A similarity between the approaches is that IPOT also includes an implicit warm-start of the dual variable, which can be considered a special case of our warm-start proposed in Section 4.2.1, where $\Delta_\gamma^{(t)} = 1$ for all $t$.

In Figure 9, we compare IPOT with MDOT ($p=1.5, q=2^{1/3}, \gamma_{\rm i}=2$) on the upsampled MNIST dataset. For a fair comparison, we followed the conventions of Xie et al. (2020) and implemented all algorithms by computing row/column sums of $P(\boldsymbol{u}, \boldsymbol{v}; \gamma)$ explicitly rather than via LogSumExp reductions as in Algorithms 2 and 4.[4] Each matrix-vector product, row/column sum, row/column scaling of a matrix and entry-wise operations on matrices is counted as one operation of cost $O(n^2)$. Since MDOT satisfies its stopping criterion (reaches $\gamma_{\rm f}$ and satisfies $\|\nabla g(\gamma_{\rm f})\|_1 \leq H_{\min}(\boldsymbol{r}, \boldsymbol{c})/\gamma^{-p}$) up to $10\times$ more quickly, we continue minimizing $g(\boldsymbol{u}, \boldsymbol{v}; \gamma, \boldsymbol{r}, \boldsymbol{c})$ using the respective KL projection algorithm (Sinkhorn or PNCG) until the same number of $O(n^2)$ operations as IPOT are executed (namely, $5 \times \gamma_{\rm f}$).

Overall, the methods behave similarly for the $L_1$ cost, while MDOT is up to $10\times$ faster for the $L_2^2$ cost.

These results also reveal an interesting contrast between the $L_1$ and $L_2^2$ cost functions. For the $L_1$ cost the optimality gap rapidly approaches 0 as the KL projection becomes more precise (see dashed lines), even

---

[4]In this implementation, we evaluate the Sinkhorn direction (17) for PNCG by adding a small constant $\approx 10^{-30}$ to each entry of $\boldsymbol{r}(P)$ and $\boldsymbol{c}(P)$ for numerical stability.

though $\gamma_f$ is fixed. This suggests that the entropic gap, namely $\langle P^*(\gamma) - P^*, C \rangle$, is small at $\gamma_f$ for these MNIST problems, and the optimality gap is dominated by the inexactness of the projection onto $\mathcal{U}(\boldsymbol{r}, \boldsymbol{c})$. We believe that the fast rate $O(\exp\{-\gamma K\})$ of Weed (2018) discussed in Section 4.2.2 is active here for the entropic gap, although this requires further study. On the other hand for the $L_2^2$ cost, continuing to iterate on the projection error to improve the approximation of $P^*(\gamma_f)$ does not reduce the optimality gap any further, which implies the entropic gap $\langle P^*(\gamma_f) - P^*, C \rangle$ is still dominant at $\gamma_f$ in these problems.

# H  Additional Benchmarking on DOTmark

Figs. 10-19 add further benchmarking on 10 more datasets from the DOTmark benchmark of Schrieber et al. (2017), which include various kinds of randomly generated images, classical test images and real data from microscopy. Each dataset contains 45 unique pairs of marginals $(\boldsymbol{r}, \boldsymbol{c})$ obtained from pixel values. The cost matrix is constructed from distances in 2D pixel locations; we evaluate on both $L_1$ and $L_2$ distance costs for $n = 4096$ following our setup in Sec. 5.

Besides clock time, we additionally plot here the total number of $O(n^2)$-costing operations for each algorithm, e.g., matrix-vector products, row/column sums of matrices, vector outer products, element-wise operations on matrices. We count primitive operations for consistency across algorithms; counting a higher-level function call such as the number of gradient evaluations would be unfair due to inherent differences in the design of various algorithms. For instance, some require costly line search between gradient evaluations. These plots show that operation counts of the baseline algorithms follow similar trends to wall-clock time and no algorithm is unfairly advantaged via low-level optimizations.

For each of 20 problem sets (10 image datasets × 2 cost functions), 20 out of 45 problems are sampled without replacement. The wall-clock time plots for the respective cost functions ($L_1$ and $L_2^2$) follow similar trends as Fig. 5. In addition to the median, we also include 75% confidence intervals along both axes, which show that MDOT is generally robust.

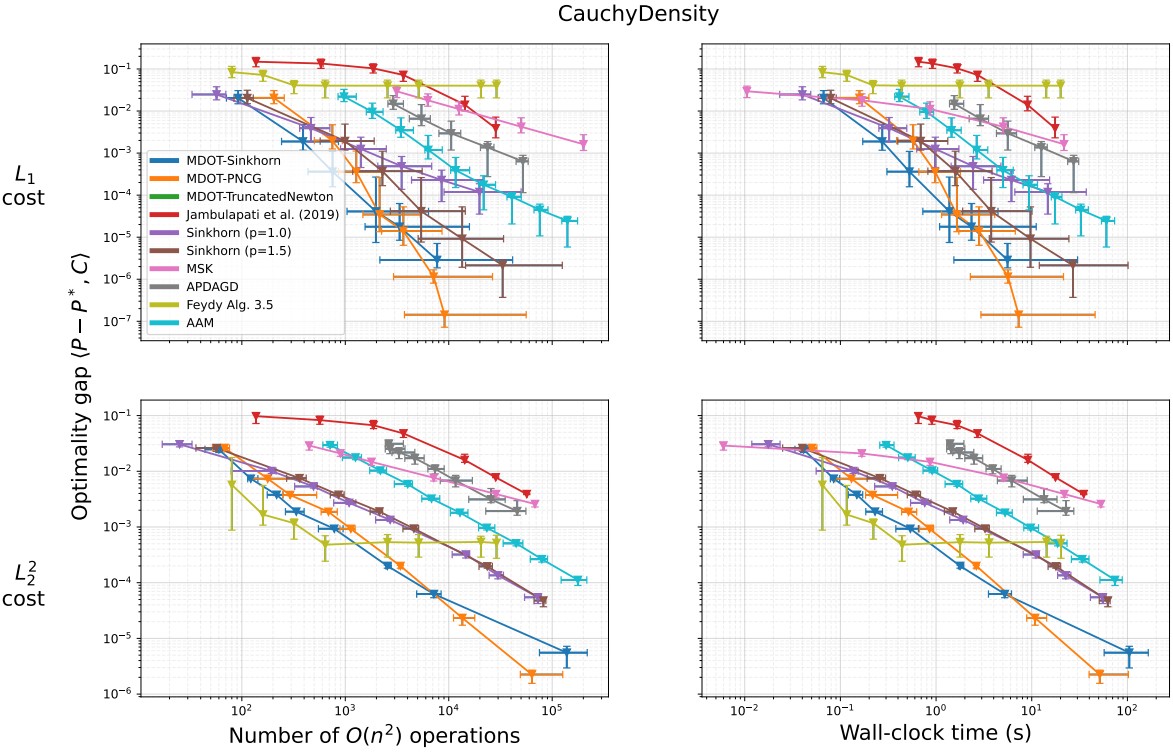

Figure 10: `CauchyDensity` problem with $L_1$ (top) and $L_2^2$ (bottom) costs, showing excess cost (error) vs. number of $O(n^2)$ operations (left) and wall-clock time (right).

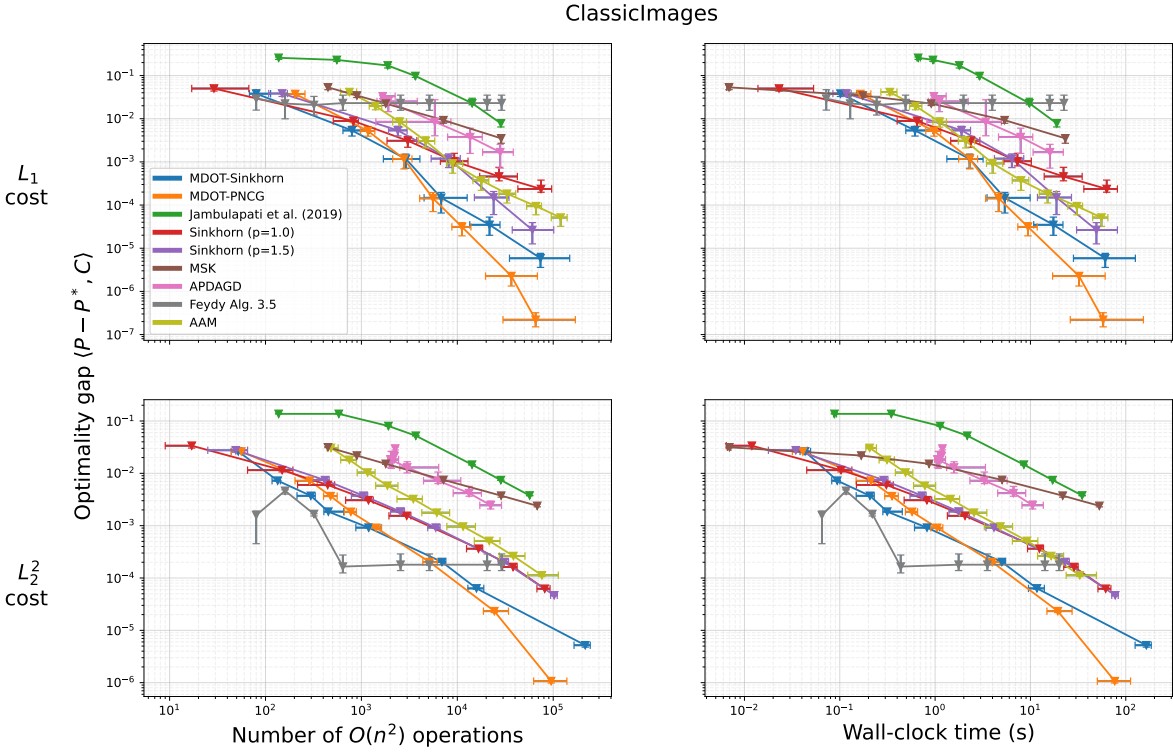

Figure 11: `ClassicImage` problem with $L_1$ (top) and $L_2^2$ (bottom) costs, showing excess cost (error) vs. number of $O(n^2)$ operations (left) and wall-clock time (right).

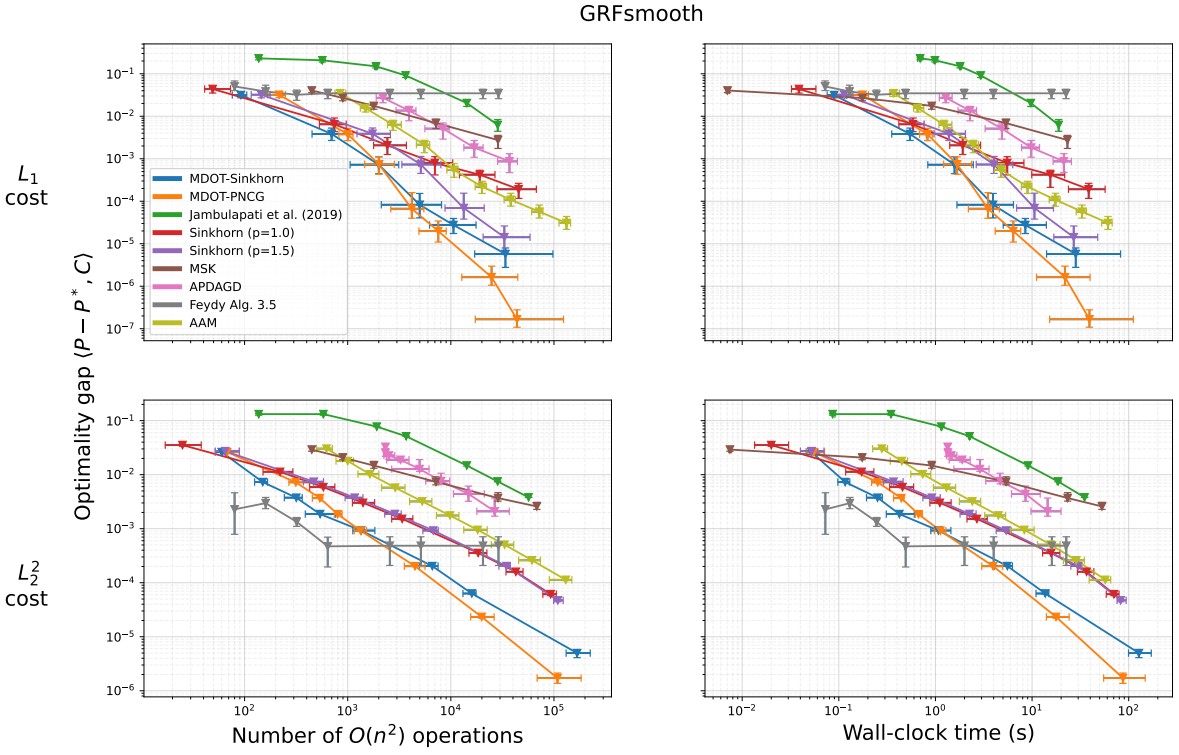

Figure 12: `GRFSmooth` problem with $L_1$ (top) and $L_2^2$ (bottom) costs, showing excess cost (error) vs. number of $O(n^2)$ operations (left) and wall-clock time (right).

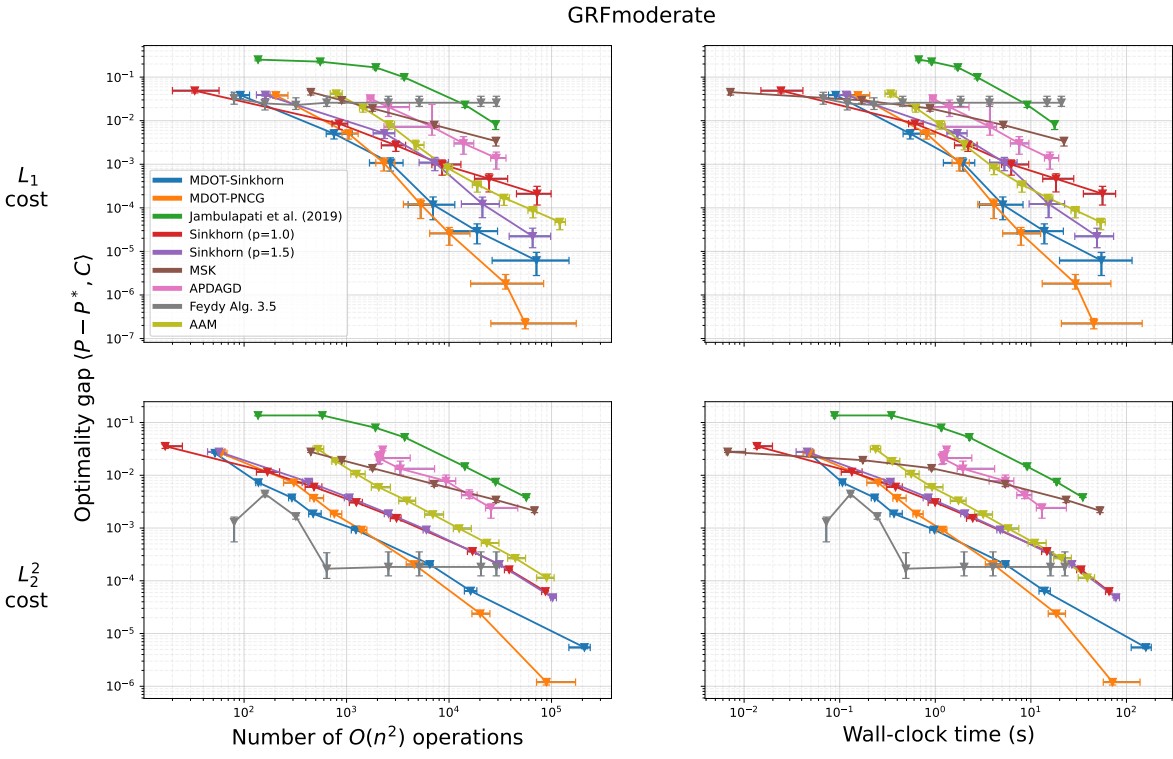

Figure 13: `GRFModerate` problem with $L_1$ (top) and $L_2^2$ (bottom) costs, showing excess cost (error) vs. number of $O(n^2)$ operations (left) and wall-clock time (right).

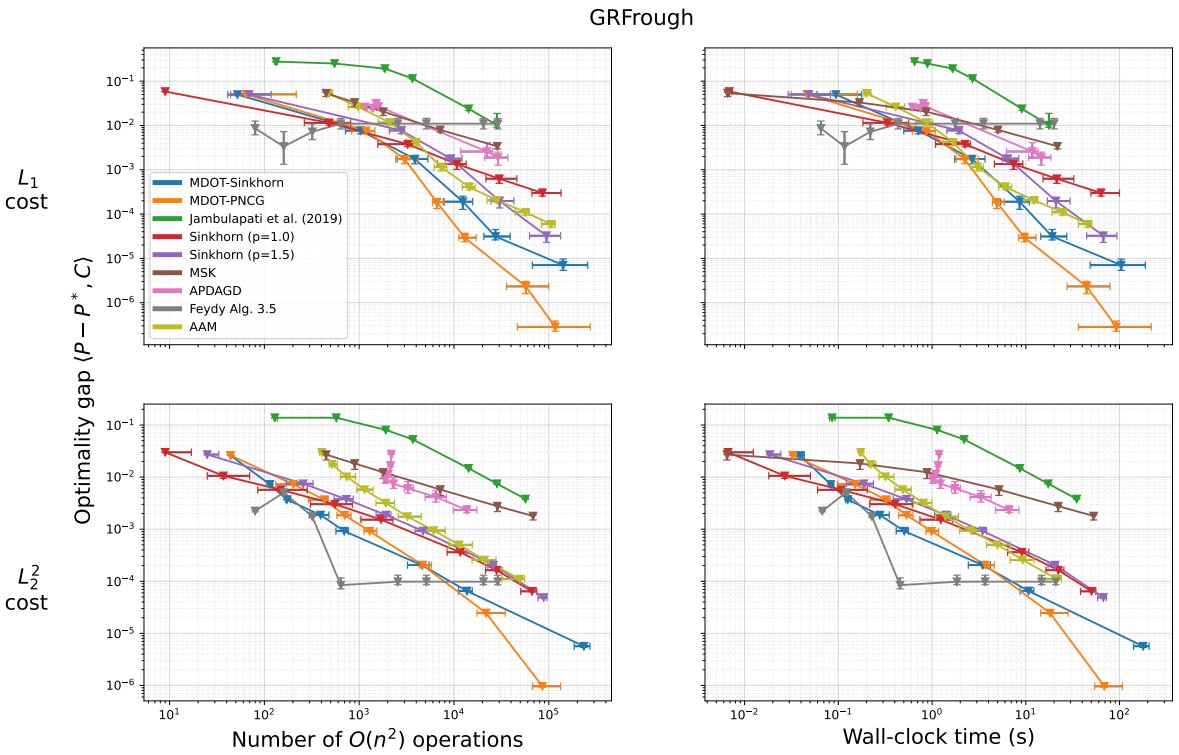

Figure 14: `GRFRough` problem with $L_1$ (top) and $L_2^2$ (bottom) costs, showing excess cost (error) vs. number of $O(n^2)$ operations (left) and wall-clock time (right).

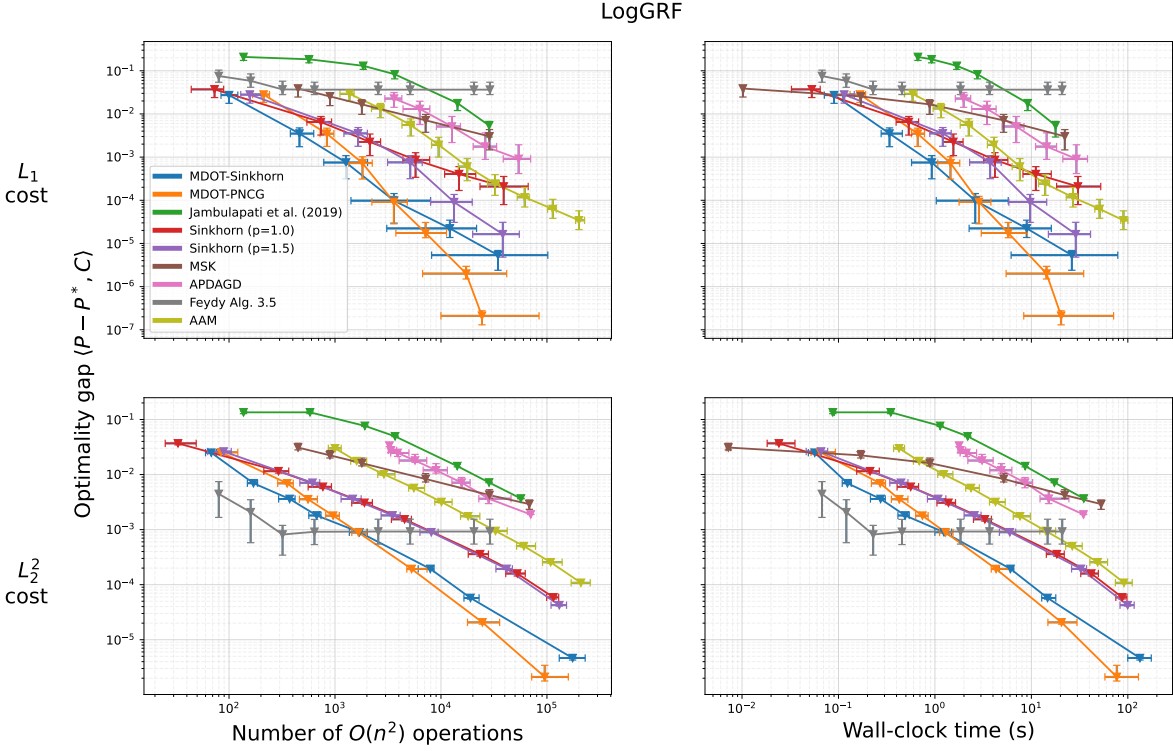

Figure 15: `LogGRF` problem with $L_1$ (top) and $L_2^2$ (bottom) costs, showing excess cost (error) vs. number of $O(n^2)$ operations (left) and wall-clock time (right).

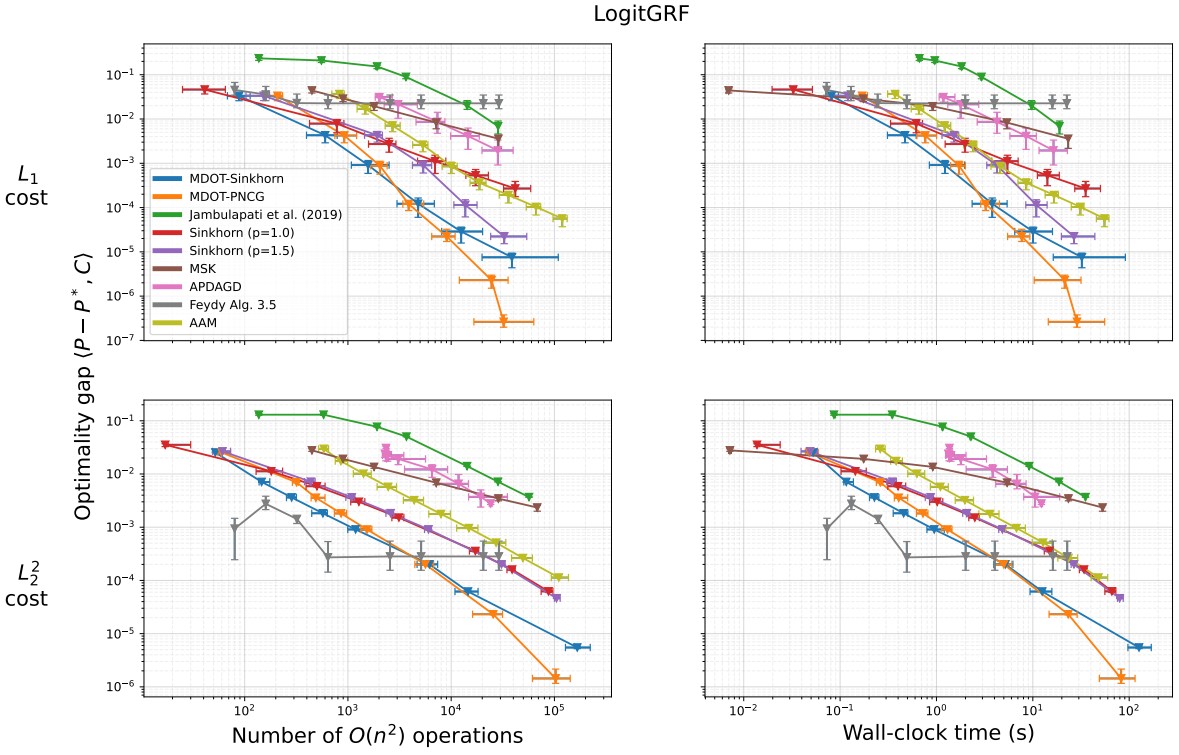

Figure 16: `LogitGRF` problem with $L_1$ (top) and $L_2^2$ (bottom) costs, showing excess cost (error) vs. number of $O(n^2)$ operations (left) and wall-clock time (right).

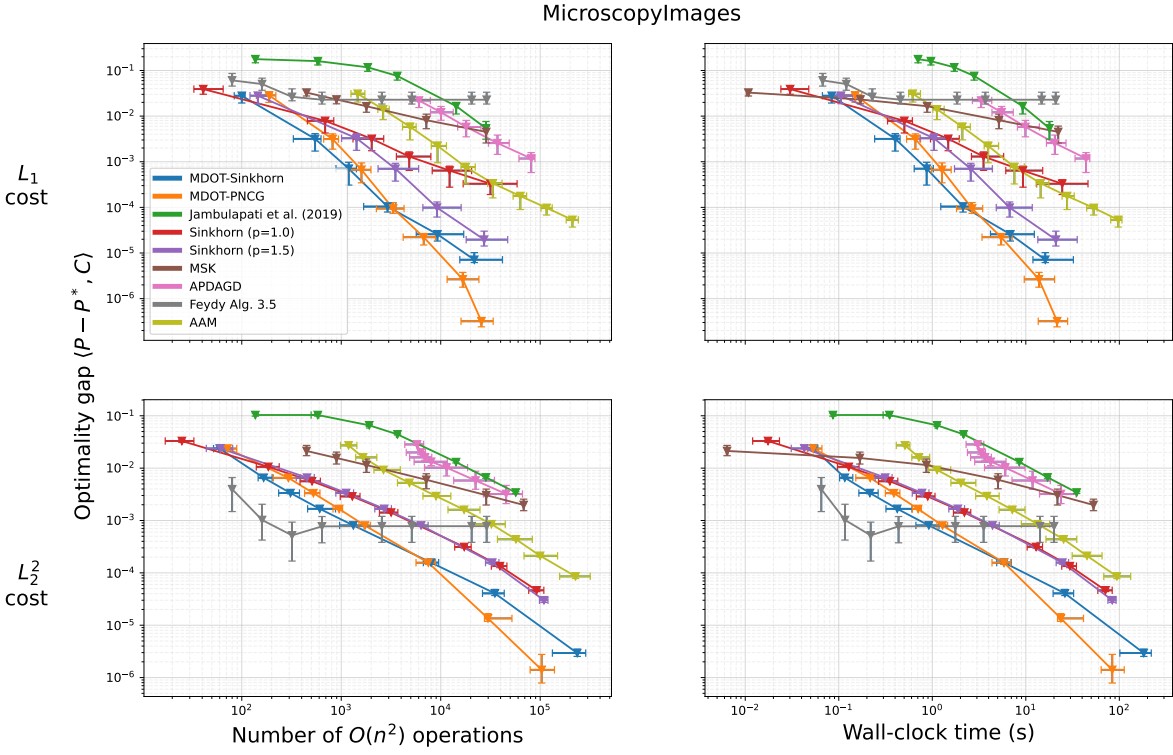

Figure 17: `MicroscopyImage` problem with $L_1$ (top) and $L_2^2$ (bottom) costs, showing excess cost (error) vs. number of $O(n^2)$ operations (left) and wall-clock time (right).

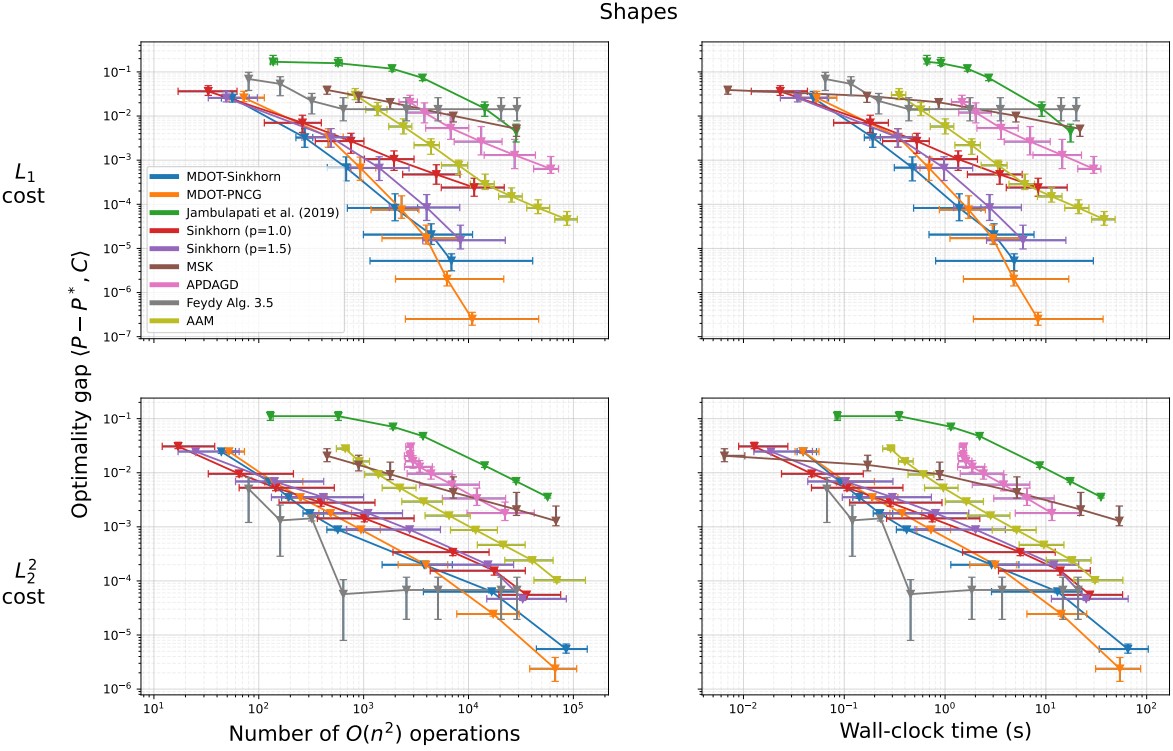

Figure 18: `Shape` problem with $L_1$ (top) and $L_2^2$ (bottom) costs, showing excess cost (error) vs. number of $O(n^2)$ operations (left) and wall-clock time (right).

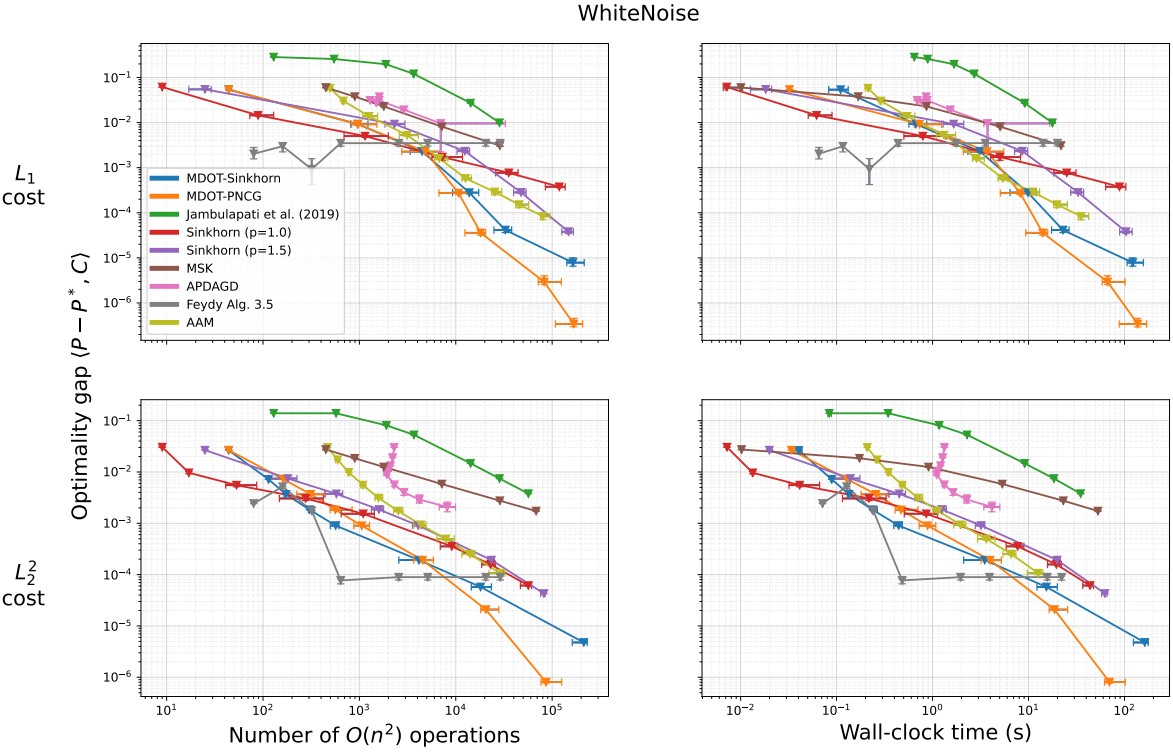

Figure 19: `WhiteNoise` problem with $L_1$ (top) and $L_2^2$ (bottom) costs, showing excess cost (error) vs. number of $O(n^2)$ operations (left) and wall-clock time (right).

