# OpenReview forum: "Efficient and Accurate Optimal Transport with Mirror Descent and Conjugate Gradients"
_TMLR — Accepted by TMLR_

### Review · Reviewer_yvDj · 2025-03-07

**Summary Of Contributions:**

This work proposes Mirror Descent Optimal Transport for discrete OT, which applies mirror descent steps on the unregularized primal objective instead of the dual regularized objective. Instead of a projection given by Sinkhorn iterations, a Bregman projection is used instead, computed approximately using a conjugate gradients method. Experiments demonstrate significantly improved memory footprint and faster convergence compared to existing OT methods, as well as experimental support for the scaling properties.

**Audience:**

Yes

**Broader Impact Concerns:**

Usage of generative AI models to generate data for color transfer experiment.

**Claims And Evidence:**

Yes

**Requested Changes:**

**Critical**
* Address weaknesses, especially proof of Prop 4.

**Strengthening**
* (Fig 3) Change the caption or the figure to point out that the y-axis is in terms of the threshold $\epsilon_d$.
* Big-O notation is used inconsistently as $\tilde{O}$ earlier and $O$ later (e.g. section 5).
* Add short takeaways to the captions of each figure.

[1] Gramfort, A., Peyré, G., & Cuturi, M. (2015). Fast optimal transport averaging of neuroimaging data.
[2] Bonet, C., Nadjahi, K., Sejourne, T., Fatras, K., & Courty, N. (2024). Slicing Unbalanced Optimal Transport.

**Strengths And Weaknesses:**

**Strengths**
* The work is generally easy to read.
* The performance of the proposed algorithm is compelling and is compared with many other algorithms. There is also a suitable selection of experimental verifications of the scaling properties of the proposed method.

**Weaknesses**
* Using "exponentiated gradient descent" (mirror descent with entropy mirror map) in place of regular gradient steps has been done in [1] in the context of computing Wasserstein barycenters. Please add this as a reference and possible other applications of mirror descent in the context of optimal transport in the literature review.
* The introduction to mirror descent in Sec 2.2 is short and confusing. In particular, the $P$ notation and the new introduction of $f$ which is only defined at the end of this section. Please rewrite in a more general manner.
* (Critical) Prop 4.1 seems sketchy, as the statement is very surprising, in particular that the single MD step can work to exactly solve the regularized OT problem from any rank-1 initialization. I believe the issue is in the recursion step (below Eq. 23, there is sum over $t$, up to $t$...). I think it is fine to simply remove the erroneous claims, including Prop 4.2. It is worth checking Prop 4.3 as well as it appears not to refer to the initialization (again, surprising). Prop 4.4 seems plausible but I did not check it as the statement does not look useful.
* Limited theoretical properties of the proposed algorithm, e.g. convergence rates. It also does not link into the proposed algorithmic components, which are plentiful and motivated but poorly linked to content earlier in the paper.
* How is $P^*$ computed in Figure 4?
* Using the entropic mirror descent has the distinct disadvantage of not being able to reach the boundary of $\mathbb{R}_{\ge 0}$ in the mirror descent step. How does the proposed method compare with competing methods when the true transportation plan lies on the boundary, like if the cost matrix has zero entries off the diagonal? This appears to be a possible numerical instability problem with the proposed Sinkhorn direction Eq. 18.
* Why is the unregularized objective being compared e.g. in Figure 2 when the target problem is the regularized objective? How do the regularized objectives compare? It would be interesting to see how the regularized OT objective compares instead of the unregularized primal, as the competing methods solve the entropic OT problem. Perhaps the slow convergence of the other methods is due to solving an approximate problem, in which case, the proposed MDOT problem has a clear advantage.

---

> ### Author Response · Authors · 2025-03-27
> **Responses and the new revision (1/2)**
>
> We thank the reviewer for their time and constructive feedback, as well as their appreciation of the significance of our empirical results and readability.
>
> **Exponentiated Gradient Descent (Mirror Descent) Related Work:**
> Section 3 (paragraph 2) discussed relevant prior works involving exponentiated gradient descent or mirror descent ideas. Following your suggestion, we have now added the reference [Gramfort et al. (2015)] to the end of this paragraph to contextualize our work more broadly. Additionally, we've reorganized Section 4.2 (up to Sec. 4.2.1) to clarify precisely how MDOT differs from prior methods employing similar concepts.
>
> **Clarity and Improved Flow in Background Section 2.2:** We revised and expanded Section 2.2, introducing mirror descent more intuitively and clearly. A new Equation (7) now concludes this section, providing a smoother transition to the main contributions in Section 4.
>
> **Proposition 4.1 and Proof Typo:**
>
> We recognize that Proposition 4.1 (now Proposition 4.2) previously appeared surprising. To clarify:
> - We reorganized Section 4.1 to simplify exposition and incrementally build intuition, introducing a Lemma 4.1 before Proposition 4.2.
> - The previous proof of the first statement of the Proposition had a typo; the summation index was intended to run from t′=0 to t. We have now revised the exposition in our proof entirely to be shorter and more direct, with a lot of the technicality moved into the proof of Lemma 4.1. The revised exposition, both in the main text and in the proofs in the Appendix, should clarify the intuition and correctness of these statements.
> - The discussion following the proposition now explicitly connects to the construction of MDOT detailed in Section 4.2 via Eq. (10), which we additionally illustrate visually in a newly added Figure 1.
>
> **Lack of Theoretical Convergence Rate:** We agree that the lack of a theoretical convergence rate of the proposed algorithm is a limitation of this work, but underline that (i) we make no claims regarding a theoretical convergence rate, but rather point to it as interesting future work and (ii) to address this gap, we presented extensive empirical results across 24 problem sets (12 datasets × 2 costs), thoroughly benchmarking against 7 baselines across multiple hyperparameter settings (Figures 5, and 10–19). Such comprehensive empirical validation goes beyond what is typical for similar papers (e.g., Altschuler et al., 2017; Dvurechensky et al., 2018; Lin et al., 2019; Jambulapati et al., 2019; Guminov et al., 2021; Ballu & Berthet, 2023; Luo et al., 2023; Tang et al., 2024).
>
> **Exact Computation of $P^\*$**: $P^*$ is computed by running an exact CPU-based solver from the Python Optimal Transport (POT) package. We have clarified this in Section 5 (end of 1st paragraph).
>
> **Behavior at Large $\gamma$ and Boundary Contact:** Indeed, strictly positive solutions are a natural consequence of entropic regularization, and exact boundary contact only occurs in the infinite limit. Nonetheless, as $\gamma \rightarrow \infty$, small entries quickly vanish (below numerical precision limit), effectively converging to boundary solutions in practice. We test MDOT up to a large value of $\gamma=2^{18}$.
>
> **Numerical Stability of Equation 17 (previously 18):** The log-space operation of Equation 17 sets it apart from Equation 16 (previously 17) and provides numerical stability as studied in Sec. 4.3. Thanks to the use of Sinkhorn direction (Equation 17) and a careful design of the line search procedure (Appx. B.2), we are able to implement the PNCG algorithm exclusively using [stabilized LogSumExp reductions](https://gregorygundersen.com/blog/2020/02/09/log-sum-exp/). The use of stabilized LogSumExp reductions is now repeated immediately below Eq. (17). Alternatively, simply adding a tiny constant of order $10^{-30}$ to $\mathbf{r}(P)$ and $\mathbf{c}(P)$ can stabilize a more direct implementation (this was done to generate results in the newly added Appx. F).

---

> > ### Author Response · Authors · 2025-03-27
> > **Responses and the new revision**
> >
> > **Why is the unregularized objective being compared e.g. in Figure 3 (previously Fig. 2) when the target problem is the regularized objective?**
> >
> > The target problem in our work is actually the unregularized objective (which we address through the EOT objective under weak regularization). We leverage entropic regularization as a numerical tool to approximate the unregularized OT solution by progressively weakening regularization (i.e., increasing $\gamma$). Our updated Section 4.1 clarifies this procedure. Specifically, Figure 3 (left) demonstrates that tuning the parameter $p$ (introduced in our work) improves the empirical convergence rate toward the unregularized solution for Sinkhorn iteration, aligning with our theoretical insights in Section 4.2.2. Our overall approach builds on and extends previous work on accelerating convergence to the unregularized solution using entropic regularization, by increasing $\gamma$ (either progressively or by tuning) and improved dual optimization algorithms (see Altschuler et al., 2017; Dvurechensky et al., 2018; Lin et al., 2019; Guminov et al., 2021).
> >
> > **How do the regularized objectives compare?**
> >
> > Recall that by minimizing the EOT dual objective $g$, we are approaching the regularized primal from below and we have equality (up to a linear transformation at optimality; see Sec. 2.2 of Lin et al., 2019). Hence, convergence measured by the dual gradient norm $||\nabla g||_1$, as shown in Figure 4 (previously Fig. 3), directly indicates convergence of the regularized primal objective. Figure 4 specifically highlights that our PNCG algorithm achieves faster convergence compared to Sinkhorn iterations, particularly under weaker regularization (large $\gamma$). MDOT-PNCG exploits this faster convergence in subsequent results (Figures 5, 10–19) to better approximate solutions to the unregularized OT problem compared to Sinkhorn and MDOT-Sinkhorn.
> >
> > **Strengthening Suggestions:** We agree and will incorporate the requested figure caption clarification, consistent Big-O notation, and concise figure takeaways in the camera-ready version.
> >
> > **Broader Impact Concern:** We have provided supplementary materials containing all generative AI images used for color transfer experiments, confirming their harmless nature.

---

> > > ### Comment · Reviewer_yvDj · 2025-03-28
> > > **Response to authors**
> > >
> > > I thank the authors for their revisions to the manuscript and the detailed responses to my questions, which have addressed my concerns. I will edit my recommendation accordingly.
> > >
> > > **Minor**
> > > * Titles of sections 5 and 6 are in all capitals.

---

### Review · Reviewer_7ogQ · 2025-03-09

**Summary Of Contributions:**

The paper introduces Mirror Descent Optimal Transport, which unifies temperature annealing in entropic-regularized OT with mirror descent techniques, for solving the optimal transport problem approximately.

**Audience:**

Yes

**Claims And Evidence:**

No

**Requested Changes:**

Please address the above-mentioned and the following issues.

To improve clarity, the authors should provide a more comprehensive and structured derivation of the algorithm from the mirror descent framework. This should include a clear explanation of the theoretical underpinnings, assumptions, and any approximations made. Furthermore, the authors should ensure that the context is self-contained, allowing readers to follow the development of the algorithm without needing to consult numerous external sources.

Moreover, while the idea of using mirror descent (or Bregman/Entropic proximal methods) for solving optimal transport is not novel (see e.g., [1]), the paper should better highlight the advanced techniques and innovations in the proposed method compared to existing methods. This could include a detailed comparison of the proposed algorithm's performance, complexity, and convergence properties relative to other state-of-the-art methods. By doing so, the authors can more effectively demonstrate the significance and impact of their contributions.

[1] Xie, Y., Wang, X., Wang, R., and Zha, H. A fast proximal point method for computing exact Wasserstein distance. In Uncertainty in artificial intelligence, pp. 433–453. PMLR, 2020.

**Strengths And Weaknesses:**

# Strength

The authors propose a GPU-parallel nonlinear conjugate gradients algorithm to efficiently solve the subproblem, comparable to existing algorithms such as Sinkhorn iterations. The effectiveness and efficientcy of the proposed method are demonstrated on various datasets.

# Weaknesses

The paper is not very easy to follow. In particular, the connection between the main algorithm, i.e., Algorithm 1, and the mirror descent framework is not clear based on the current presentation. The explanation lacks a detailed step-by-step derivation that would help readers understand how the algorithm is formulated from the mirror descent principles. Additionally, the paper cites many references when discussing the main algorithm, but the information is presented in an unstructured manner, making it difficult to discern the key contributions and innovations.

---

> ### Author Response · Authors · 2025-03-27
> **Responses and the new revision**
>
> Thank you for your constructive feedback. We have revised the manuscript in light of your comments.
>
> **Clarification of Algorithm 1 from Mirror Descent Framework:**
>
> We substantially revised Sections 4.1 and 4.2 (up to Section 4.2.1) to clarify how Algorithm 1 (MDOT) follows directly from the mirror descent framework. In addition:
> - We introduced a new Figure 1 to visually illustrate the conceptual and mathematical relationships clearly.
> - The previously opaque “BregmanProject” step in Algorithm 1 has been replaced with a clear and direct minimization step (Line 6), explicitly stating the objective $g$ and the convergence criterion ($L_1$ gradient norm).
> - We have replaced the term “Bregman projection” with “KL projection” everywhere else where applicable, since this is the only kind of Bregman projection we use in practice in this work.
> - Algorithm 1 now appears immediately below its illustrative figure, accompanied by a concise line-by-line summary of each step with links to the mirror descent derivation.
>
> We believe these revisions substantially improve clarity and allow the reader to clearly follow the MDOT algorithm’s derivation from mirror descent principles in a self-contained manner. The construction of the PNCG algorithm is based on non-linear conjugate gradients (NCG), which has an extensive literature.  While we provide a primer on CG and NCG in Section 4.3, a more extensive coverage is outside the scope of this paper.
>
> **Structured Comparison with Existing Methods:**
>
> We reorganized and expanded Section 4.2 to explicitly and systematically highlight how each component of MDOT differs from or improves upon existing methods employing mirror descent-type approaches. Each key innovation or novel aspect is now clearly contrasted with prior work (e.g., Xie et al. 2020, already cited previously in Related Work). To clarify specifically with respect to IPOT (Xie et al. 2020), we've further added an extended discussion and empirical data to Appx. F (see Fig. 9).
>
> The reader is referred to this section Appx. F (3rd line of Page 7), after a short summary of how MDOT differs from IPOT: unlike MDOT, IPOT does not necessitate a bound on the the dual gradient norm, but rather it only takes a fixed, small number of L Sinkhorn iterations (usually L=1) after each temperature update. They update the inverse temperature $\gamma$ in fixed increments (typically, $1$), while we leverage extrapolated warm-starting to increase $\gamma$ more aggressively. This allows us to reach the extremely weak regularization regime more quickly (shown in Fig. 9). Note also, IPOT only allows Sinkhorn updates between temperature updates, while MDOT may benefit from arbitrary (possibly better) convex optimization algorithms such as PNCG.
>
> **Comparison of the proposed algorithm's performance, complexity, and convergence properties to baselines:**
>
> In Figures 5, and 10–19, we already compared the algorithm’s performance and convergence properties empirically against 7 existing OT methods across extensive hyperparameter configurations on 24 distinct problem sets (12 datasets × 2 cost functions, 20 problems per configuration) in terms of both wall-clock time and number of O(n^2) operations. While we left a theoretical convergence rate analysis for future work, (i) we did not claim a theoretical rate in this paper, (ii) to address this gap, we presented extensive empirical benchmarking beyond what is typical for similar papers (e.g., Altschuler et al., 2017; Dvurechensky et al., 2018; Lin et al., 2019; Jambulapati et al., 2019; Xie et al., 2020; Guminov et al., 2021; Ballu & Berthet, 2023; Luo et al., 2023; Tang et al., 2024).

---

> > ### Comment · Reviewer_7ogQ · 2025-03-28
> > **Follow Up**
> >
> > The author has generally addressed my concerns. However, I noticed some potentially inappropriate comments regarding the development of inexactness conditions for related algorithmic frameworks. These conditions are crucial because, in practice, the subproblem cannot always be solved exactly. It is also essential to carefully consider the domain of the kernel function that defines the Bregman distance. Specifically, in the convergence analysis, when computing the Bregman distance $D(x,y)$, the point $y$ must lie in the interior of the domain of the underlying kernel function. This requirement helps explain why existing works introduce complexity conditions to ensure the validity of their analysis and avoid inaccuracies. Could the authors address this issue?

---

> ### Author Response · Authors · 2025-03-30
> **Minor revision**
>
> Thank you for pointing out this consideration. To address this concern, we revised Sections 3 and 4.2 of our manuscript to clarify more precisely the differences between our approach and Yang & Toh (2022). We note that since all MDOT iterates share the structure $P(\mathbf{u}, \mathbf{v}; \gamma) = \exp \\{ \mathbf{u}\mathbf{1}^\top + \mathbf{1}\mathbf{v}^\top - \gamma C \\} $, they naturally remain in the interior of the domain of the kernel function (negative Shannon entropy). Please let us know if any further concerns remain.

---

> > ### Author Response · Authors · 2025-04-01
> > **Following up**
> >
> > Dear Reviewer, we submitted our revised manuscript and response to address your comments a few days ago. As we near the end of the response period, we wanted to check in to see if there are any remaining questions or further clarifications needed from our side.

---

> > > ### Comment · Reviewer_7ogQ · 2025-04-01
> > >
> > > Thank you. I don’t have any other comments. I will change my rating later.

---

### Review · Reviewer_Xten · 2025-03-18

**Summary Of Contributions:**

The work proposes a method called Mirror Descent Optimal Transport (MDOT), which empirically produces superior performance to existing OT algorithms in the discrete setting. It works by progressively decreasing the regularization for entropic OT problems in a fashion motivated by mirror descent applied to the primal form of the problem. Several practically impactful tweaks to existing methods are presented:
* a different tolerance $\varepsilon_d$ from [Dvuruchensky et al. 18], [Lin et al. 19], and [Altschuler et al. 17]
* a warm start to the Bregman projection different from that of [Schmitzer 19] and [Feydy 20]
* use of a pre-conditioned nonlinear conjugate gradients method for Bregman projections instead of the standard Sinkhorn-Knopp iterations

They demonstrate on a set of problems from upsampled MNIST dataset and a color transfer task and show that their method empirically outperforms existing methods in terms of LogSumExp calls and wall-clock time, especially when higher precision is required.

**Audience:**

Yes

**Broader Impact Concerns:**

No concerns here.

**Claims And Evidence:**

Yes

**Requested Changes:**

I do not have too many requests, as I was not able to check the article in great detail. One small note I have is that it would be helpful to place some lines on Fig. 5 (e.g., $n^{2.5}$) to guide the estimations referenced in the text. It's hard to see otherwise (especially with varied scaling on the horizontal and vertical axes).

**Strengths And Weaknesses:**

Strengths:
* The empirical performance of the method is strong.
* I felt the writing was good, and communicated the basic idea of the method quite well.
* There appear to be well-reasoned motivations behind their tweaks to the method.

Weaknesses:
* Most evidence is empirical for the superiority of their method and there is not much in the way of proven guarantees on improved performance.

---

> ### Author Response · Authors · 2025-03-28
> **Thank you for your feedback.**
>
> We appreciate the reviewer’s positive assessment.
>
> We acknowledge that our results primarily emphasize empirical evidence; exploring theoretical performance guarantees is indeed an interesting direction for future research, but is beyond the current manuscript.
>
> Regarding the suggested improvement to Figure 5, we agree that including reference lines (such as slope indicators) will aid in interpreting the empirical scaling behavior. We will incorporate this suggestion into a revised version of the figure (camera ready).
>
> Thank you for the constructive feedback.

---

### Decision · Action_Editor_Vc1y · 2025-04-30

**Recommendation:** Accept with minor revision

**Comment:**

The paper was appreciated by the reviewers that still had some concerns in particular about the proof of Prop 4.1,  the positioning wrt existing OT solvers and the formulation as a Mirror Descent.

The responses and the new revision have convinced all three reviewer that the paper is OK for an Accept and I concur. But a few things remain to be done in the final version:
- The authors promised to add to Figure 5 lines illustrating the stated "empirical speed" which is indeed necessary because it is not clear form the Figure.
- The authors stated in their reply to yvDj that the focus in Fig 3 is to solve exact OT. This is an interesting reply and the proposed method indeed seem better at that than competitors. But SOTA solvers for exact OT are not entropic/Sinkhorn based, they are network simplex solvers. The paper need to discuss and compare to the exact OT solver (they already use the one from POT) in term of complexity and computational time. For instance $10^4$ logsumexp reduction of complexity $N^2$ for $N=4096$ might not compare favorably to the network flow whose complexity which is $N^3log(N)$ (runs in 2 seconds on my laptop at this size). Admittedly comparison between GPU and CPU solver is a bit unfair toward CPU but would also illustrate the point if the network flow is competitive.  Note that whatever methods is faster is not really important but since the focus is exact OT loss. But this discussion/paragraph needs to be in the paper is order to bring concrete numbers for future users of the solver especially since the "empirical" complexity stated in the abstract are very interesting but potentially misleading for practical problems. For instance in Figure 5 the authors must add a black vertical line with the wall-clock time of the exact solver.

**Audience:**

This paper and theme is definitely of interest to the TMLR audience.

**Claims And Evidence:**

The claims and evidence are well supported in the revised version.